computational biology, ecology, health and disease and epidemiology

disease vector, mechanistic modelling, spatio-temporal dynamics, experimental and field data, mortality scenario

**Author for correspondence:**
Pauline Ezanno
e-mail: pauline.ezanno@inrae.fr

# Dispersal in heterogeneous environments drives population dynamics and control of tsetse flies

Hélène Cecilia[1], Sandie Arnoux[1], Sébastien Picault[1], Ahmadou Dicko[2], Momar Talla Seck[3], Baba Sall[4], Mireille Bassène[3], Marc Vreysen[5], Soumaïla Pagabeleguem[6,7], Augustin Bancé[8], Jérémy Bouyer[2,5,9] and Pauline Ezanno[1]

[1]INRAE, Oniris, BIOEPAR, 44300 Nantes, France
[2]Cirad, INRAE, ASTRE, University of Montpellier, Montpellier, France
[3]Institut Sénégalais de Recherches Agricoles, Laboratoire National d'Elevage et de Recherches Vétérinaires, Dakar-Hann, Senegal
[4]Direction des Services vétérinaires, Ministère de l'Elevage et des Productions animales, Sphères ministérielles de Diamniadio, Bât. C, 3ème étage, Senegal
[5]Insect Pest Control Laboratory, Joint FAO/IAEA Programme of Nuclear Techniques in Food and Agriculture, 1400 Vienna, Austria
[6]Insectarium de Bobo-Dioulasso – Campagne d'Eradication des Tsé-tsé et Trypanosomoses (IBD-CETT), Bobo-Dioulasso 01, BP 1087, Burkina Faso
[7]Université de Dédougou (UDDG), BP 176, Burkina Faso
[8]Centre International de Recherche-Développement sur l'Elevage en Zone Subhumide (CIRDES), Bobo-Dioulasso 01 01 BP 454, Burkina Faso
[9]UMR 'Interactions hôtes-vecteurs-parasites-environnement dans les maladies tropicales négligées dues aux trypanosomatides', Cirad, Montpellier, France

HC, 0000-0001-5687-3570; SP, 0000-0001-9029-0555; PE, 0000-0002-0034-8950

Spatio-temporally heterogeneous environments may lead to unexpected population dynamics. Knowledge is needed on local properties favouring population resilience at large scale. For pathogen vectors, such as tsetse flies transmitting human and animal African trypanosomosis, this is crucial to target management strategies. We developed a mechanistic spatio-temporal model of the age-structured population dynamics of tsetse flies, parametrized with field and laboratory data. It accounts for density- and temperature-dependence. The studied environment is heterogeneous, fragmented and dispersal is suitability-driven. We confirmed that temperature and adult mortality have a strong impact on tsetse populations. When homogeneously increasing adult mortality, control was less effective and induced faster population recovery in the coldest and temperature-stable locations, creating refuges. To optimally select locations to control, we assessed the potential impact of treating them and their contribution to the whole population. This heterogeneous control induced a similar population decrease, with more dispersed individuals. Control efficacy was no longer related to temperature. Dispersal was responsible for refuges at the interface between controlled and uncontrolled zones, where resurgence after control was very high. The early identification of refuges, which could jeopardize control efforts, is crucial. We recommend baseline data collection to characterize the ecosystem before implementing any measures.

## 1. Introduction

Environmental heterogeneity drives insect population dynamics [1]. Spatially, it induces movements from source patches (i.e. area where the local population increases) to sink patches (i.e. area where the local population decreases),

possibly enhancing population resilience in unsuitable patches [2]. Temporally, environmental suitability can vary over time both locally owing to microclimate variations, e.g. impacting vegetation growth [3], and at a larger scale owing to seasonal unfavourable periods. Spatial and temporal environmental heterogeneities, therefore, have distinct effects. If confounded, this can lead to erroneous predictions of ecological processes [4]. Relating such a complex time- and space-varying habitat with population dynamics remains a challenge [5].

This becomes crucial when managing vector-borne diseases whose transmission depends on interactions between vectors and hosts, and thus on landscape structure [6]. A rescue effect (i.e. new individuals arriving rescue local population from extinction) may occur because of spatial heterogeneity and individual dispersal, inducing resilience of the controlled population, especially if control is not area-wide (i.e. targeting an entire vector population within a circumscribed area) [7]. In addition, vectors are also subjected to seasonal habitat suitability [8]. Such spatial and temporal variations could induce unexpected changes in vector population dynamics. However, management programmes of vector populations often are designed without considering local environmental characteristics.

Tsetse flies (*Glossina* spp.) are vectors of human and animal African trypanosomes, a major pathological constraint for productive livestock and sustainable agricultural development in sub-Saharan Africa [9]. They are widely distributed across sub-Saharan Africa, occurring in 38 countries and infesting around 10 million km$^2$ [10]. Over 60 million people are continuously exposed to the risk of becoming infected, and farmers in tsetse-infested areas suffer up to 20–40% losses in livestock productivity leading to an estimated annual loss of US$4500 million [11]. Among the 31 species and subspecies of tsetse flies, less than one-third is of economic and human health importance, the remaining mostly thriving in wildlife areas or located in thinly populated forested areas [12]. Management efforts have been ongoing for decades in Africa but largely failed to create sustainable tsetse-free areas, i.e. the tsetse distribution has been reduced with less than 2% [13,14]. Although tsetse flies have a complex ecology and biology, their slow reproduction rates make them an ideal target for an eradication strategy. This, however, requires a better understanding of their spatio-temporal dynamics [15].

Modelling helps to better understand insect population dynamics [16] and predict their trends under changing conditions [17]. Sparse and heterogeneous knowledge can be integrated, producing simulations complementary to field observations and experiments [18]. Complex behaviours can be studied and scenarios tested [19], and when hypotheses and limits are clear [20], models can guide decision-making [21].

There are a few models which deal with tsetse biology and population dynamics [22–25]. They have rarely been used to guide decision-making in operational programmes [26–29]. Those which have, mostly failed to predict population resilience, leading to inaccurate guidelines [15]. In addition, most control programmes were not implemented following area-wide principles [7]. Their failure could be explained by population resurgence in non-treated patches or re-invasion by neighbouring populations [30,31]. It is still unclear which patch properties are relevant to define sources and sinks in a hostile environment created by eradication efforts, and whether dispersal can lead to population resurgence. Spatial complexity considerably influences model predictions [31–33], and population dynamics differ among local patches of variable suitability, possibly affecting dynamics at a larger scale.

Our objective was to assess the effect of spatial dispersal along with spatial and temporal environmental heterogeneity on tsetse fly population dynamics and control. Control was implemented as spatially targeted increases in adult fly mortality. We developed a mechanistic spatio-temporal model that incorporates environmental heterogeneity through a data-driven approach. The model was applied to a *Glossina palpalis gambiensis* population of the Niayes (Senegal) that is subject to an ongoing eradication project [34]. Less than 4% of the habitat is considered favourable for *G. p. gambiensis* [35] and tsetse populations are highly structured across the metapopulation [36]. This knowledge was incorporated in the model, accounting for combined effects of spatial complexity, density-dependence and temperature on the age-structured population.

## 2. Material and methods

### (a) Key knowledge on tsetse biology

Meteorological variables influence the abundance and spatio-temporal distribution of tsetse flies [37,38], with average temperature being the most influential one [39]. However, its influence compared to, or combined with, demographic processes is poorly understood. Tsetse flies reproduce by adenotrophic viviparity (electronic supplementary material, figure S1A). The egg hatches in the female's uterus. The developing larva is nourished by the milk glands until larviposition. Between 20 and 30°C, the lower the temperature is, the longer the period between larvipositions [40]. Similarly, colder temperatures prolong the pupal period [41]. The first larviposition occurs around day 18 post emergence. The period between larvipositions is 10 days on average [39]. Temperatures above 30°C increase adult mortality [37]. Mortality, related to predation and feeding success, is density- [42] and age-dependent [43], with remarkably high losses in nulliparous flies (up to her first larviposition) partly owing to starvation risk [39]. This species acquires feeding preferences, i.e. the host selected for the first blood meal, can influence the one selected for the second meal. This learning capability increases the hunting efficiency of older flies [44].

Tsetse flies are classified into three groups based on their behaviour, habitat preference and distributions, i.e. forest (subgenus *Fusca*), savannah (subgenus *Morsitans*) and riverine flies (subgenus *Palpalis*). While most previous models applied to the savannah species *Glossina pallidipes* and *Glossina morsitans*, we focused on the riverine species *G. p. gambiensis* that thrives in forest galleries and riparian thickets [45]. The habitat of this species stretches along rivers and, therefore, its dispersal is mostly in one dimension. However, in some areas, like the Niayes of Senegal (electronic supplementary material, figure S1B), rivers and associated vegetation have disappeared owing to climate change and the distribution of *G. p. gambiensis* populations is fragmented because of the very patchy vegetation [35]. Consequently, the populations disperse in two dimensions like tsetse flies of the *fusca* and *morsitans* groups. In addition, in such fragmented landscapes, tsetse flies display localized, small subpopulations with relatively short dispersal. Isolated populations in fragmented habitats are ideal targets for eradication using area-wide integrated pest management (AW-IPM) approaches [7,46]. Hence, our case study is of broad relevance to better understand and predict riverine tsetse fly spatio-temporal population dynamics in rapidly changing ecosystems which are gradually becoming the norm [47].

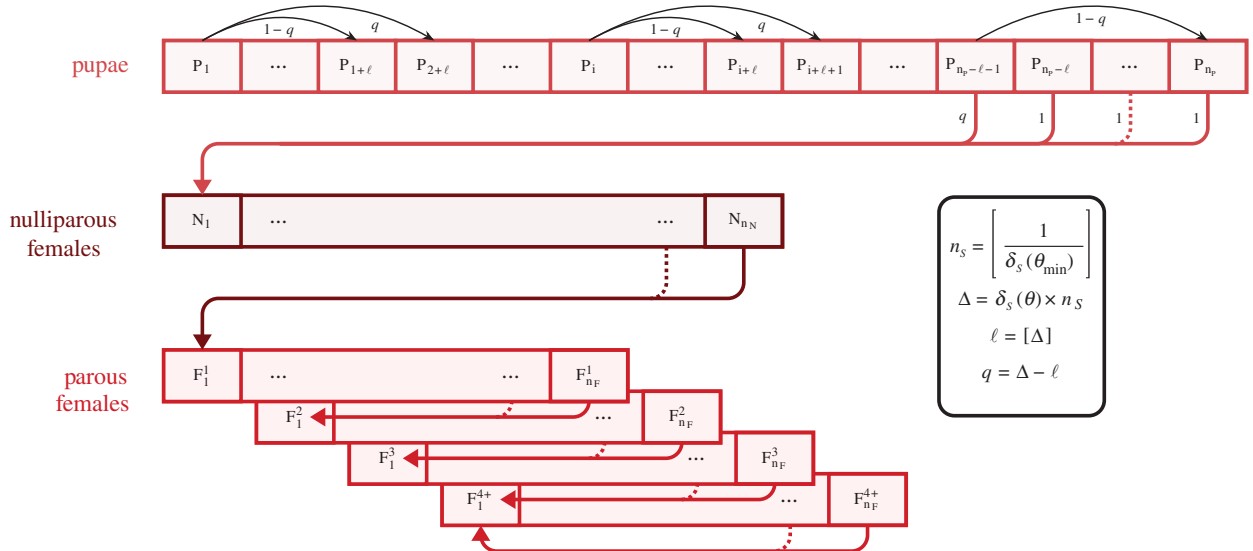

**Figure 1.** Within-cell model diagram of tsetse fly population dynamics (time unit: day). Transitions between stages except from pupa (P) to nulliparous female (N) trigger the birth of a new pupa $P_1$. Transitions occur at development rate $\delta_S$ for stage $S \in \{P, N, F_x\}$ (parity $x \in \{1, 2, 3, 4+\}$) according to temperature $\theta_{t,c}$ at time $t$ in cell $c$, giving rise daily to a minimum jump of $l$ states from each state $i$ of stage $S$, with $(1-q)S_{t,c,i}$ individuals going from state $S_{t,c,i}$ to state $S_{t+1,c,i+l}$ and $qS_{t,c,i}$ individuals going to $S_{t+1,c,i+l+1}$. If $i + l > n_S$ (respectively, $i + l + 1 > n_S$, with $n_S$ the maximum number of states in stage $S \in \{P, N, F_x\}$), then selected individuals go to the next stage. Equations are in the electronic supplementary material, S4.1. (Online version in colour.)

## (b) Data on tsetse biology and the environment

The effect of temperature on mortality, fecundity and the length of the pupal period of *G. p. gambiensis* was assessed under experimental conditions for the specific populations inhabiting the target area [48] (electronic supplementary material, S2.1). Dispersal of *G. p. gambiensis* was assessed from release–recapture data [49] (electronic supplementary material, S2.2). Field data on population age structure were available thanks to a study focusing on natural abortion rates (electronic supplementary material, S2.3). The observed age structure was compared with simulation results for qualitative evaluation of the model. In tsetse, the physiological age structure is an important feature which depends mainly on the survival rate and the length of the pupation period, both conditioned by temperature [12]. When the temperature drops (cold dry season), pupation lengthens, adult flies live longer and the population becomes older [41]. The reverse occurs during the hot dry season.

The spatio-temporal environmental heterogeneity was realistically represented using a data-driven approach. First, local carrying capacities (electronic supplementary material, figure S1C and S3.1) were defined as the maximum sustainable number of tsetse flies per cell (model spatial unit), estimated as SI × ADT/σ. SI is the suitability index (estimated with a species distribution model [34] based on maximum entropy), σ the trap efficiency (probability that a trap catches a fly within 1 km² within a day [50], set to 0.003 [34]) and ADT the apparent density of the population (flies sampled per trap per day) [51]. All available trap catch data collected in the Niayes before the start of the eradication campaign (2007–2010) were used. Second, local daily temperatures were modelled by approximating local temperatures truly perceived by tsetse flies. Temperatures measured in weather stations are not those experienced by flies in resting places which are normally 2–6°C cooler than ambient temperature [52]. Furthermore, temperature increases from the centre to edges of gallery forests. High resolution macro-climate data freely available for 2011 in the studied area were corrected using temperature data recorded in selected suitable habitats (electronic supplementary material, S3.2).

## (c) A mechanistic spatio-temporal model of tsetse fly population dynamics

A mechanistic and deterministic compartmental model was developed to predict the spatio-temporal dynamics of the

*G. p. gambiensis* population accounting for environmental heterogeneity and density-dependence. The most relevant data available for *G. p. gambiensis* (published and new) were used to calibrate the equations representing the processes involved in the population dynamics, i.e. adult mortality, pupal development and time to larviposition. Locally, individuals were categorized into pupae (P), without differentiating males and females, nulliparous females (N) and parous females with four ovarian ages ($F_1$, $F_2$, $F_3$, $F_{4+}$, figure 1) [53]. Adult male (M) densities are considered not limiting for mating. However, they play a role in density-dependent processes and were thus modelled with only death affecting their number [12]. We considered the flies not limited in their access to hosts.

The environment was modelled using a grid (cell resolution: 250 × 250 m; study area: 30 × 30 cells; electronic supplementary material, figure S1C). The model, fully described in electronic supplementary material, S4, was implemented in Python as a discrete-time model with a one-day time step (code available, electronic supplementary material, S7). Parameter values are provided in electronic supplementary material, table S2.

Within each cell, the population size per life stage decreased with mortality. We used a constant mortality rate for pupae, given the lack of data on this parameter [27]. Mortality of nulliparous females was twice as high as that of parous females [54]. Adult mortality increased with temperature above 24°C, while below this threshold and for the range of temperatures observed in the field, a constant mortality was assumed [33,39]. Density-dependence occurred when the adult population exceeded the carrying capacity of a cell [39]. In addition, individuals evolved between stages as a function of temperature. Pupal development was fitted on data (electronic supplementary material, S2.1). For nulliparous and parous females, consistency of experimental data was checked against published equations [39]. A pupa was produced at the end of both nulliparous and parous female stages.

Fly dispersal pattern was designed to favour suitable over unsuitable habitat (electronic supplementary material, S4.2). A sigmoidal density-dependent dispersal rate was assumed [55], adapted for individuals competing to access resources [42]. Fly dispersal to neighbouring cells was determined by the relative cell attractiveness, designed to favour the emptiest cells and cells of greatest carrying capacity if similarly filled. We used data to calibrate the dispersal radius, i.e. the maximum discrete jump (in

number of cells) allowed in a day (electronic supplementary material, figure S4).

After a 3-year burn-in period starting from local populations at carrying capacities and using reference parameter values (electronic supplementary material, table S2), a scenario without control was implemented over 3 years and analysed (electronic supplementary material, S4.3, S4.4). Carrying capacities were spatially heterogeneous but assumed constant over time. Daily perceived temperatures were estimated per cell for 1 year and were repeated the following years. The model integrated all current knowledge on processes governing population dynamics of *G. p. gambiensis*. We end up with a single model structure, where each biological process was carefully calibrated based on new or published data on our study species. However, as parameter calibration cannot be perfect, the individual and joint effects of parameter variations on aggregated output variance (electronic supplementary material, S5), and especially population size, were evaluated. We performed a variance-based global sensitivity analysis using the Fourier amplitude sensitivity testing (FAST) method, recommended for this type of model [56]. Predicted age structure was compared with field data for females of ovarian age 1, 2 and 3, because traps do not catch nulliparous and females of ovarian age 4 and 4+ as efficiently as females of intermediate ovarian ages [57].

### (d) Spatially targeted control strategies and population resurgence

Optimized strategies were defined by the minimal proportion of cells to be controlled, their optimal location and the control effort required. The methods used are fully described in the electronic supplementary material, S6. In brief, control was mimicked by increasing female mortality during 1 year, starting from the same initial conditions as in the scenario without control. For iteratively reduced proportions of controlled cells (starting with a homogeneous control), we assessed the minimal mortality increase required to reduce the female population size to 5 and 2% of its initial value over the whole grid after 1 year. We stopped reducing the proportion of controlled cells once it became impossible to achieve the targeted population reduction whatever the mortality. The control efficacy was assessed with respect to the female population size at both grid and cell scales, identifying: (i) the cells which contributed most to the total female population after 1 year of control; (ii) the local impact of 1 year of increased mortality (ratio between the female abundance after 1 year in the scenario with and without control); and (iii) the cells which contributed most to population recovery (highest local growth rate) in a scenario with female mortality set back to its reference value (i.e. control has stopped) for 1 year, testing population resurgence. We analysed the relationship between the local environmental variables (carrying capacity, mean temperature, temperature variance in each cell) and the three indicators aforementioned, reflecting different properties of the population spatial structure. To assess whether spatial dispersal impacted control efficacy and population resurgence, we ran similar simulations with the dispersal process turned off.

## 3. Results

### (a) New insights from biological data

Using all available data, the female fly mortality function differed from published ones (electronic supplementary material, figure S2A). Up to 24°C, female fly mortality rate was 0.013 d$^{-1}$, but mortality increased exponentially with temperatures to reach 0.023 d$^{-1}$ at 32°C (lifespan of 43–77 days). Male mortality was higher than female mortality at all temperatures (electronic supplementary material, figure S3). Pupa emergence followed a logistic function, providing a new pattern compared to Hargrove's equation [39] (electronic supplementary material, figure S2B). The daily dispersal range proved to be less than 250 m, which was equivalent to the width of one cell (electronic supplementary material, figure S4). Finally, carrying capacities were highly heterogeneous and ranged between 106 and 104 761 flies km$^{-2}$; (median: 2330, cell resolution: 0.0625 km$^2$). By contrast, spatial variations of local temperatures were small (electronic supplementary material, figure S8).

### (b) Temperature and mortality as key factors driving population size

The scenario without control was closely in line with field observations made before the start of the eradication campaign in the Niayes (electronic supplementary material, figure S6). Population dynamics was seasonally influenced and driven by temperature as expected and variations were larger between cells than within cells (electronic supplementary material, figure S7). The population growth rate was −0.63% during the last year. On average, 40, 33 and 27% of the young parous females deposited 1, 2 or 3 larvae, respectively. The spatial variability of age structure was three to four times lower than its temporal variability.

A 5% variation in temperature resulted in significant population increase or extinction, largely outweighing the effect of a similar variation in carrying capacity (electronic supplementary material, figure S9), which emphasized the need for considering reasonable temperature variations. Model outputs other than age structure were highly sensitive to variations in adult mortality ($\mu_{\{N,F,M\}}$), which explained more than 50% of population size variance (electronic supplementary material, figure S11). It emphasized the need to study in more detail the impact of mortality variations on the spatio-temporal population dynamics. Model outputs were only moderately sensitive to variations in pupae development ($\delta_P$). Nulliparous ($\delta_N$) and parous ($\delta_F$) female development rates, pupae mortality ($\mu_P$), carrying capacities ($k$) and dispersal ($g$) barely contributed to output variance.

### (c) Efficacy of control measures driven by environmental heterogeneity and dispersal

Increasing adult mortality to levels comparable to those obtained during control programmes [26] induced a sharp decline in population size after 1 year of control (figure 2). The life expectancy of the female flies had to be reduced from 60 (no control) to 35 days, to reduce the population to 2% of its original size using a homogeneous control effort (point '2', figure 2a). Cells of highest carrying capacity contributed the most to population size irrespective of whether control was implemented (figure 2b2) or not (figure 2b1). At low fly population density (66 flies km$^{-2}$), new patterns emerged related to cell-specific properties. Surprisingly, increasing mortality homogeneously had a heterogeneous impact: the decrease in local relative population density (i.e. the local control efficacy, figure 2c2) was not correlated with carrying capacity (electronic supplementary material, figure S12A), but with local temperature. The coldest cells that experienced the smallest variations in temperature showed the least impact (electronic supplementary material, figure

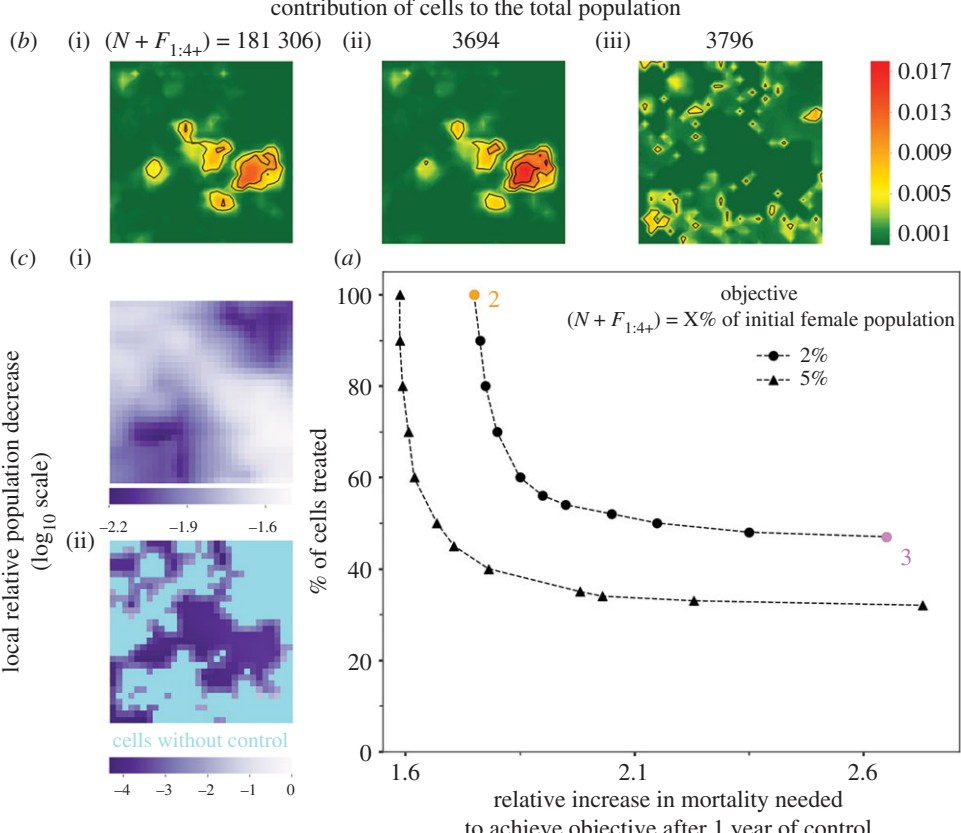

**Figure 2.** Mortality increase and tsetse fly population size. (*a*) Relative increase in mortality needed to reduce the female population size to 2% (circle) or 5% (triangle) of its initial size after 1 year of control, when a fraction of cells was targeted. (*b*) Contribution of cells to the total population size ((i): no control, (ii): homogeneous control, (iii): heterogeneous control targeting 47% of the cells). (*c*) Local control efficacy ((i)–(ii): same as in *b*), the darkest being the most effective (lighter colour: uncontrolled cells). (Online version in colour.)

S12B–D). This pattern was obvious despite the small variations in mean temperature (23.7–24.3°C) and standard deviation (1.98–2.37°C), which were not correlated (electronic supplementary material, figure S12C and D). When neglecting tsetse fly dispersal, results were similar but with a lower effect of temperature and less variations among cells (electronic supplementary material, SI7, figures S14 and S15).

Targeting cells contributing the most to population management (greatest carrying capacity and most impacted by an increased mortality) could achieve a similar decrease in population size as a homogeneous control, but required an additional effort in increasing mortality in targeted cells. Besides, the population was much more fragmented and control efficacy was no longer related with temperature. Controlling 70% of well-chosen cells was as efficient as controlling the entire area (almost the same mortality was applied). Reducing further the proportion of controlled cells required a sharp increase in mortality to obtain a similar efficacy. To reduce population to 2% of its original size while applying a heterogeneous control ( pink point '3', figure 2*a*), life expectancy had to be reduced from 60 (no control) to 23 days in 47% of the cells (average life expectancy over the entire area of 43 days). Controlling less than 47% of the cells could not reduce population below 2% of its initial size. Cells contributing the most to the total population were scattered in the area (figure 2*b*(iii)). The local relative population decrease (figure 2*c*(ii)) was slightly associated with carrying capacity (electronic supplementary material, figure S13A) and the control was more effective in cells with intermediate than low carrying capacity. No effect of temperature was observed

(electronic supplementary material, figure S13B,C). Similar patterns were observed if the population was to be reduced to 5% of its initial size. By contrast, if neglecting fly dispersal, the control efficacy was much less variable. Temperature and its variations impacted control efficacy as in the homogeneous case (electronic supplementary material, figure S16). This indicated that a non-spatial model largely misestimates the efficacy of targeted control and may lead to different conclusions.

## (d) Population resurgence after control

Population resurgence 1 year after control did not lead to reach pre-control population size, irrespective of whether control was homogeneous or heterogeneous, but resurgence could be very high in local refuges. After a homogeneous control effort, the population growth rate was 34.7% yr$^{-1}$ at the grid scale and was highly heterogeneous in space (figure 3*a*). The highest and positive growth rates were found in refuges, i.e. the coldest cells with the least variation in temperature (figure 3*c,d*) and where the impact of the control effort was previously the lowest (brown symbols). One year after the control effort had stopped, local growth rates were still negative in cells where the control had been effective (green and blue symbols). Carrying capacity did not impact resurgence (figure 3*b*). Similar results were obtained when neglecting fly dispersal (electronic supplementary material, figure S17).

In the case of a heterogeneous control effort, the growth rate of the population was lower (13% yr$^{-1}$ at grid scale) but very high in a few refuge cells where the population could

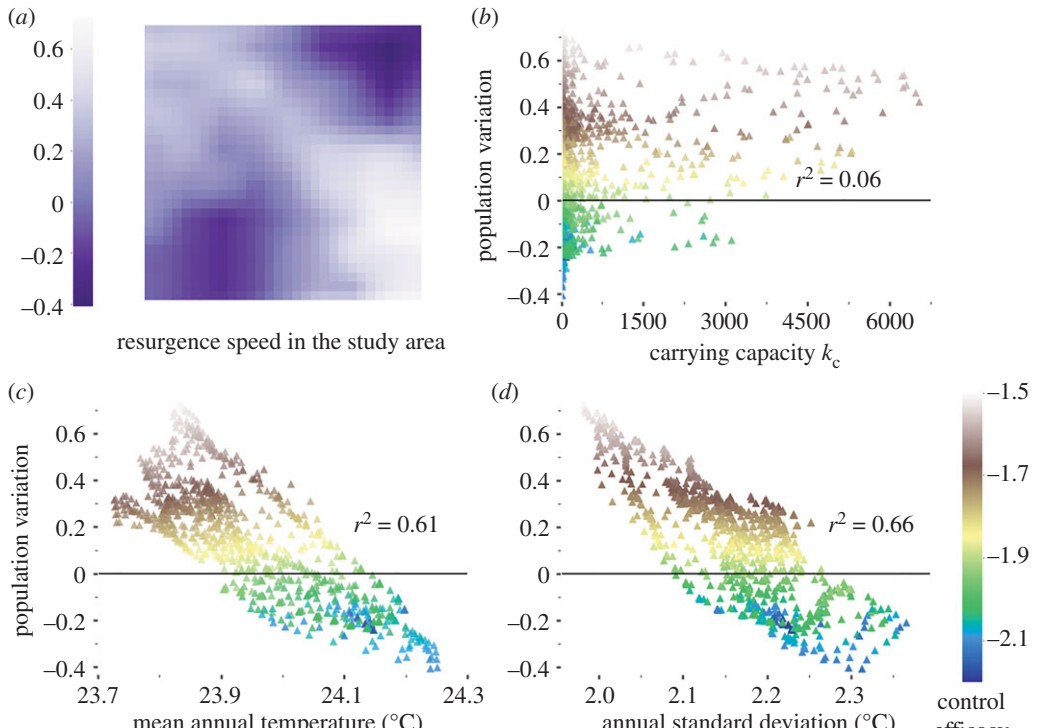

**Figure 3.** Local population resurgence 1 year after the end of a homogeneous control. (*a*) Local growth rate, (*b*–*d*) variations of the local growth rate with carrying capacity, mean annual temperature and annual standard deviation of temperature. Colours: control efficacy (blue (bottom colour): most effective). (Online version in colour.)

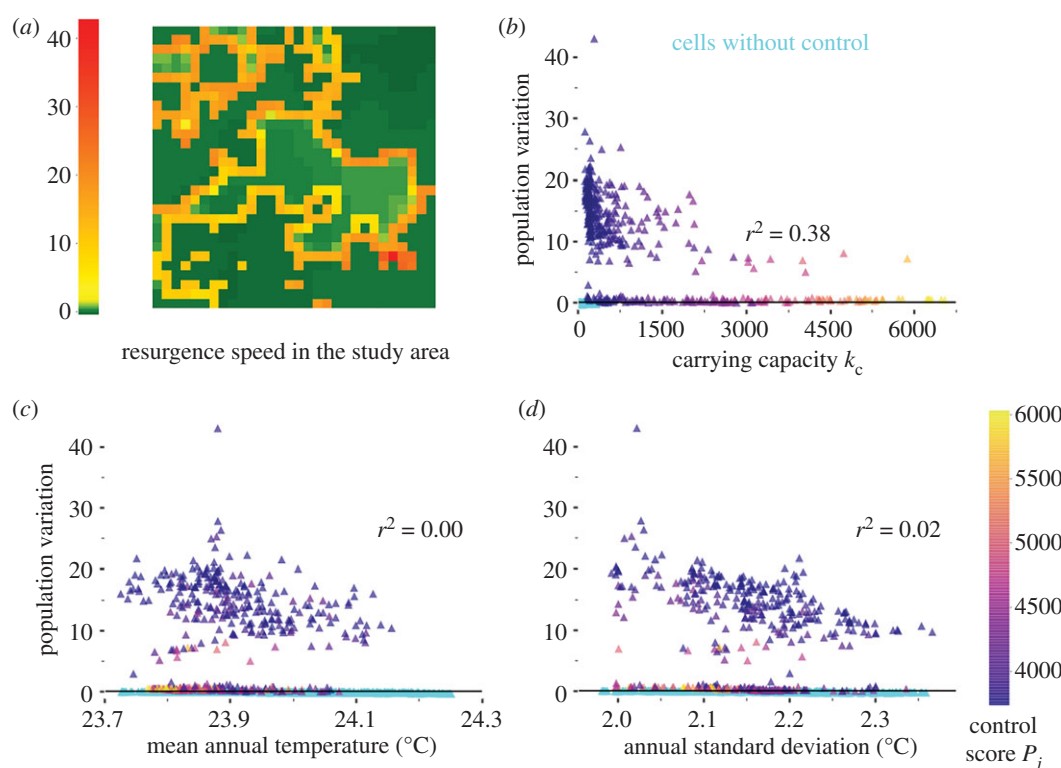

**Figure 4.** Local population resurgence 1 year after the end of a heterogeneous control (47% of controlled cells). (*a*) Local growth rate, (*b*–*d*) variations of the local growth rate with carrying capacity, mean annual temperature and annual standard deviation of temperature. Colours: control score (the highest: still targeted when the proportion of controlled cells is reduced). Cyan: uncontrolled cells. (Online version in colour.)

be multiplied by 6–44 (figure 4*a*). It was not correlated with local characteristics or with scores used to target controlled cells (figure 4*b*–*d*). Refuges were located on the interface between controlled and uncontrolled zones (figure 4*a*). Monitoring efforts after the control period should particularly focus on cells of intermediate carrying capacity (figure 4*b*). Results

largely differed when neglecting dispersal (i.e. no recolonization is possible), results being then similar to the homogeneous case (electronic supplementary material, figure S18). This shows again that using a non-spatial model is not relevant to assess targeted control strategies and subsequent consequences on population dynamics.

# 4. Discussion

Environmental heterogeneity, together with spatial dispersal of individuals, not only drives the temporal dynamics of *G. p. gambiensis* populations at a large scale, but also its spatial distribution and control efficacy. We showed that a homogeneous increase in adult mortality resulted in a heterogeneous impact. The coldest cells that had the smallest variations in temperature acted as refuges, where the control was less effective and the population resurged faster after the control effort was stopped. We propose that, to increase the chances of success, control strategies should account for environmental heterogeneity and emphasize: (i) local areas of high suitability characterized by a high carrying capacity, (ii) local refuges characterized by lower local temperatures within the relevant range for tsetse (23.7–24.0°C here), and (iii) local areas with low variability of the temperature. In the case of a heterogeneous and optimized control, targeting the most relevant cells resulted in more dispersed individuals, and control efficacy was no longer related to temperature. This could be owing to the score used to target cells, which is a balance between cell contribution and control impact. Indeed, cells preferentially targeted are the ones where the population decreases the most during control, especially if they highly contribute to the global population. Population resurgence was slow after the heterogeneous control effort had stopped, but was very high in local refuges that were located at the interface between controlled and uncontrolled zones. Refuges, despite representing only a small surface area suitable for tsetse flies, could jeopardize control efforts by providing areas from which recolonization may occur after control, reinforcing the need for area-wide integrated pest management approaches [58]. We also evidenced the need for spatial models to assess targeted control strategies and their consequences on the population dynamics after control has stopped. We favoured spatial heterogeneity over temporal variability of control measures because it is closer to what is observed in the field. In a control programme, insecticide traps are set at a heterogeneous density based on the availability of suitable habitat, and then maintained and replaced when necessary (at least every six months) to maintain the control pressure [34]. Likewise, the density of sterile males to be released is proportional to the amount of suitable habitat and constant over time.

This mechanistic spatio-temporal model developed to predict the dynamics of *G. p. gambiensis* populations and how these behave when adult mortality is increased is innovative compared to already published models. Our model incorporated environmental heterogeneity through a data-driven approach and accounted for variable temperatures and carrying capacities in space and time, as well as for fly dispersal. It can be applied to other areas with available data and a known metapopulation structure. Using a landscape predicted from remote sensing data and realistic temperature data, the importance of refuges was evidenced, which was previously not obvious. Published models barely account for the spatial dynamics of tsetse populations [27,59], or only in one dimension while assuming theoretical carrying capacities [33]. A single spatial agent-based model has been proposed previously [25], which dynamically assigns patch suitability in a binary way (1 or 0) using environmental data, but which does not account for temperature-dependence in biological processes and only tests homogeneous control

scenarios. Predicted age structure was in good agreement with field data and proved robust as it barely varied following parameter variations. Amplitude and duration of seasons are expected to be major drivers influencing ovarian age distribution of the population, but this could not be assessed here as temperature data were only available for 1 year.

In terms of model parametrization, recent field and laboratory data on mortality, development and dispersal were incorporated into our model, decreasing the paucity of information on tsetse species other than *G. m. morsitans* [60]. Our sensitivity analysis highlights the need for more biological studies to better infer mortality variation with temperature, and more accurately estimate temperatures as perceived by insects, an issue recently raised for another tsetse species [60]. Such a complementary interplay between models, field observations and laboratory experiments is fundamental to make accurate predictions.

The effect of temperature on fly population dynamics both at large and local scales emphasizes the need for further investigating the impact of climate change on tsetse populations [61]. It is unlikely that flies will cross the Sahara, but they can migrate to higher altitudes particularly in eastern and southern Africa and invade trypanosoma-free zones [12]. In addition, adjacent isolated populations could merge owing to the changing habitat, which could possibly impair control operations. Conversely, new populations could become isolated especially in view that temperature is the main driver of landscape friction for tsetse fly populations [46].

The dynamics of tsetse fly populations proved to be much more sensitive to mortality than reproduction and this is consistent with the specialist nature of tsetse flies occupying a narrow niche. Individual survival of *Glossina* spp. is prioritized over reproduction [48], as they have evolved towards optimal use of energy and resources [62]. This makes them highly adapted to their ecological niche. Therefore, tsetse flies are less likely to leave their habitat and expose themselves to other environments, which keeps the population at or near carrying capacity. The field of ecology has theorized in the past that fast action methods (e.g. insecticides) are better suited for species showing high reproductive rates, short generation times, broad food preferences and good dispersal abilities [63]; and that, in contrast, pests reproducing at low rates with long generation time, but good competitive abilities would be more efficiently managed using host resistance, cultural and sterilization methods. This has already been nuanced, as such characteristics should be considered in conjunction with species relationships in communities [63]. Our study shows that it is time to move past this dichotomy and consider the effect of combined and spatially targeted control measures to achieve eradication.

Traps, targets and insecticide-treated livestock are control tactics increasing adult mortality, which can drastically reduce tsetse populations [34,64,65]. However, disregarding the existence of refuges, where increasing mortality is not as effective as in other areas, can give a false sense of accomplishment. Indeed, obtaining very low tsetse densities can be insufficient to eradicate the population as recently demonstrated against *G. p. gambiensis* in northwestern Ghana [66], the Loos islands in Guinea [64] and the Mouhoun river in Burkina Faso [65]. The obtained decrease in population densities in those examples fluctuated between 90 and 99%, which are the levels of population reduction used in our model (95 and 98%). Heterogeneous control is feasible in

practice. For instance, traps can be installed heterogeneously over the habitat (e.g. [34]). Our model will be very useful to highlight which zones to control. An interesting perspective would be to predict how many traps per hectare should be set per zone, according to zone characteristics and the level of mortality targeted locally.

Our model provides a relevant tool to evaluate complex strategies as it accounts simultaneously for density-dependence, spatio-temporal environmental heterogeneity and all stages of the tsetse flies' life cycle possibly targeted by control measures. Our approach gives indications on how to trigger a drastic decline of the population. It provides cues on how to spatially optimize control but could further minimize the operational burden by proposing optimal periods of intervention. Future developments should include more realistic, diverse and customable control scenarios, evaluated not only based on their efficacy for population reduction but also their cost in terms of on-the-ground implementation effort. In addition, to predict population dynamics at very low densities and assess final steps of eradication strategies, a stochastic framework should be developed to enable quantifying the probability of population extinction.

To conclude, carrying capacity largely explained the contribution of local source spots to tsetse fly population dynamics at a large scale, but unfavourable conditions resulted in a progressive disappearance of such spots and the existence of refuges located in cold areas where the temperature was less variable. Areas to be controlled should be chosen with caution when facing a heterogeneous habitat. A homogeneous control effort applied for 1 year had less impact on tsetse population size in these refuges. By contrast, applying a heterogeneous control resulted in refuges located on the interface between controlled and uncontrolled zones, and previous temperature-dependent refuges disappeared. We confirmed that the study area should be characterized before control to target the most relevant cells, which is the underlying concept of AW-IPM. Fly dispersal should be accounted for to adequately assess the efficacy of such a heterogeneous control.

Data accessibility. Data and code are available in the electronic supplementary material and online: https://sourcesup.renater.fr/projects/spatial-tsetse/.

Authors' contributions. J.B. and P.E. designed the study and advised biological details. H.C., S.A., S.Pi. and P.E. developed the model. H.C. conducted the analyses and prepared the figures. H.C., S.A., S.Pi., J.B. and P.E. discussed the results. H.C. and P.E. wrote the manuscript. A.D. provided model external input data readily usable by the mechanistic model. J.B., M.T.S., B.S., M.B., M.V., S.Pa. and A.B. collected the data. All authors edited the manuscript.

Competing interests. We declare we have no competing interests.

Funding. This work has been conducted within the project 'Integrated Vector Management: innovating to improve control and reduce environmental impacts' (IVEMA) of Carnot Institute 'Livestock Industry for the Future' (F2E). This project received funding from the European Research Council under the European Union's Horizon 2020 research and innovation programme (grant agreement no. 682387—REVOLINC). The contents of this publication are the sole responsibility of the authors and do not necessarily reflect the views of the European Commission.

Acknowledgements. The authors are thankful to the technicians of the vet services from Senegal and ISRA for collecting the field data used in this study. A previous version of this paper [67] has been peer-reviewed as a preprint and recommended by Peer Community in Ecology (https://dx.doi.org/10.24072/pci.ecology.100024).

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
