## [Reviewer comments · Proceedings of the Royal Society B: Biological Sciences]

Review History

RSPB-2019-1735.R0 (Original submission)

Review form: Reviewer 1

Recommendation

Major revision is needed (please make suggestions in comments)

Scientific importance: Is the manuscript an original and important contribution to its field?

Good

General interest: Is the paper of sufficient general interest?

Acceptable

Quality of the paper: Is the overall quality of the paper suitable?

Acceptable

Is the length of the paper justified?

Yes

Should the paper be seen by a specialist statistical reviewer?

No

Do you have any concerns about statistical analyses in this paper? If so, please specify them explicitly in your report.

Yes

It is a condition of publication that authors make their supporting data, code and materials available - either as supplementary material or hosted in an external repository. Please rate, if applicable, the supporting data on the following criteria.

Is it accessible?

Yes

Is it clear?

N/A

Is it adequate?

N/A

Do you have any ethical concerns with this paper?

No

Comments to the Author

The authors of this paper develop and implement an age-structured model of tsetse fly population dynamics. They are interested in optimizing control strategies and hypothesize that failure to account for spatial variability could lead to population resurgence in unexpected areas. Failure to do area-wide interventions can lead to re-invasion by neighboring populations.

The paper is generally well written, but I think some of the writing could be revised for clarity. That is, there are some key terms which are not defined and this could make the paper hard to follow at points – e.g. cell, sources, sinks, patches, refuges. In addition, there are a number of phrases used which are unclear to me – e.g. “landscape configuration,” “rescue effect,” “grid scale.” Some sentences are unclear – see notes below.

My major concern is that the authors may have overstated some of their results. It seems like the authors had only one model with many constrained parameters. While some sensitivity analysis was performed (lines 125-6) and they do compare fits with observed age-structure, they do so only for intermediate aged flies and this model does not appear to be fit or calibrated to data; it was only stated that the population levels appeared to be close to the observed levels. While in a modeling context it's difficult to explore every possible scenario, since the authors didn't explore several model structures and/or parameter sets, it's unclear to me how sensitive their results are to the specific model assumptions and parameters they chose. I.e. two reasonably well-calibrated models can produce dramatically different results under intervention.

I'm particularly skeptical of the result that “female mortality function differed from published one” – field and experimental observations informed model inputs, so this seems somewhat inconsistent to favor certain observations over others. Furthermore, it's not clear to me based on this statement alone that field observations of mortality are flawed because a model required different mortalities.

Lastly, some minor concerns:

- The control strategies are not entirely clear to me. Were all combinations of cells tried at all levels? This seems like too many possible combinations to examine. I.e. which 70%/46% of the area was controlled? (Lines 187-8). How was this decided?
- I recommend any values should have confidence intervals – i.e. at the very least, the variability

described in the sensitivity analysis should be propagated through to final estimates. E.g. percent variance explained & graphs (e.g. figure 2A).

Minor Notes

- Abstract: Spatio-*temporal* and dynamical model seems redundant since a dynamical model must concern changes over time.
- Abstract: Spatial complexity, or spatial variability? I would change this to variability.
- Abstract: what is a cell? I gather it's a pixel on your graph in figure X?
- Introduction, lines 5-6: "Confounding the role of spatial and temporal environmental heterogeneity could induce erroneous predictions of ecological processes. Unclear. Rephrase?"
- Line 9: Replace x. I.e. write out "vector and host interactions"
- What is "landscape configuration"? "Rescue effect"?
- Clear definitions of cells, sources, sinks, and patches would be helpful.
- Lines 18-19: Revise sentences – e.g. -> They are widely distributed *across* Africa*,* *occurring* in 38 countries *and* infesting around 10 million km².
- Lines 21-22: This sentence is unclear: "Among the 31 species and subspecies of tsetse flies, a third is of economic and human health importance"
- Line 23: on-going -> ongoing
- Lines 29-30: Not sure what scanned means here.
- decision-making -> decision making
- Line 72: pest management approaches -> pest-management approaches
- Line 73: spatio-temporal population dynamics seems redundant
- Line 104: Define cell
- The list of tsetse models doesn't seem exhaustive & I am skeptical that there are so few modeling papers (e.g. Hargrove JW has a similar tsetse modeling paper from 2017)
- Lines 152-3: What is the logic in using a different pupal emergence function
- Results line 169-70: Confidence intervals on variance explained?
- Discussion: "knowledge-driven" and "data-driven" – not sure the usage here makes sense.
- Line 249 – what is meant by "binary occupancy"

Review form: Reviewer 2

Recommendation

Accept with minor revision (please list in comments)

Scientific importance: Is the manuscript an original and important contribution to its field?

Excellent

General interest: Is the paper of sufficient general interest?

Good

Quality of the paper: Is the overall quality of the paper suitable?

Good

Is the length of the paper justified?

Yes

Should the paper be seen by a specialist statistical reviewer?

No

Do you have any concerns about statistical analyses in this paper? If so, please specify them explicitly in your report.

No

It is a condition of publication that authors make their supporting data, code and materials available - either as supplementary material or hosted in an external repository. Please rate, if applicable, the supporting data on the following criteria.

Is it accessible?

N/A

Is it clear?

N/A

Is it adequate?

N/A

Do you have any ethical concerns with this paper?

No

Comments to the Author

This is a very good paper on an interesting and impactful disease, with the interaction of temperature and environmental conditions on vector borne disease being a subject of broad general importance.

I only have one more substantial comment, and this is in regards to the model validation. The authors state that partial validation was done by comparison of model age structure to field data and later that this was good and robust as it did not vary with parameters (line 244). I do find this slightly concerning, as the lack of variation would suggest it is not 'important' in the model, in the sense that other factors of importance do vary with parameters (i.e. what has been validated may be a robust quantity, but if it doesn't vary with outputs of interest as the parameters vary, how much is it telling us about the critical parts of the model). Therefore it would be helpful to identify other points of validation, preferably ones where there is greater sensitivity to parameters. Alternatively if the authors can show why age structure is a key point of validation, then that would suffice.

A further smaller point is that the proportionate area of control was chosen for an entirely rational but seemingly arbitrary reason - why is 2% of the population important (I presume the figure is used because of historical levels of control as in line 24, but this does not seem to have epidemiological importance)? If there is a reason beyond this, it would be helpful to have this clarified. Also, it seems likely that, in poorer countries, in practice this level of control of 46% in large areas seems like it might be rather high - if this is not the case, it would be worthwhile to have a brief point made in the paper. Otherwise it would seem to be worthwhile to ask the question from the point of view of a more 'practical' coverage. An interesting question in this regard given the distribution of refuges around the heterogeneous control areas would be whether or not a buffer zone (essentially a band around the edges of the top 46% area) could usefully protect spillover areas even if the total control area was smaller, given the very short daily dispersal ranges.

a few minor points follow:

line 10 should be 'and thus'

line 27 should be 'and helps us to'

line 55 is it the decrease of the temperature over the period between larvipositions that matters, or is it mean here that the period is lower at lower temperatures?

line 58 should be capital 'C' for temperatures (2x)

line 61 'to their first host'

line 101 how was the grid cell size chosen? It looks like the cell size is well below the scale at which fly population effects typically vary (at least in the figure) but it would be helpful if this relationship were quantified, e.g by something like a measure of the variation of key quantities within a cell, compared to the variation between cells. if the former is high compared to the latter, it would suggest you might be missing some important effects

line 196 the lack of importance of temperature doesn't seem intuitive to me and is therefore interesting - it would be helpful to have a bit more discussion this.

Decision letter (RSPB-2019-1735.R0)

29-Oct-2019

Dear Dr Ezanno,

I am writing to inform you that we have now received referees' reports on your manuscript RSPB-2019-1735 entitled "Environmental heterogeneity drives tsetse fly population dynamics and control". My apologies that this has taken longer than usual: we were waiting for a referee who in the end never sent in their report, so we had to find another referee.

The manuscript has, in its current form, been rejected for publication in Proceedings B. This action has been taken on the advice of the Associate Editor and the referees, who have recommended that substantial revisions are necessary. With this in mind we would be willing to consider a resubmission, provided the comments of the referees are fully addressed. However please note that this is not a provisional acceptance, and the referees raise some substantial issues: these relate partly to the presentation and the clarity of the writing, but also to the interpretation of the model.

Yours sincerely,
 Professor Loeske Kruuk
 Editor
 mailto: proceedingsb@royalsociety.org

Associate Editor
 Board Member: 1
 Comments to Author:

Combining the comments from the referees and my own reading: this paper is on an interesting and important topic, but there are a number of major issues with the manuscript. In summary of these:

1) There are issues with clarity of the paper, in particular terms not defined clearly enough, ref 1 refers to some.

2) In many places the writing is not easy to decipher, e.g. sentences in L61-62 :

Learning capability of older flies makes them return on their first host, increasing their hunting efficiency.

(Why this example is not clear: Maybe there's a grammar error ("on" rather than "in"), but also what is "learning capability". And specifically returning to the first host, presumably to feed? And no explanation of why that increases their hunting efficiency.)

3) Too much key information is left to the SI. In particular, it is not possible to deduce from the main text the general form of the model used, and this is important given it is the heart of what was done.

4) Ref 1 raised that the author's results may be overstated: specifically sensitivity analysis was limited, and no check of sensitivity to model choice. I would agree with this. See Ref 1's comments for details.

5) Related to 4) and maybe 2)... In many places there were remarks that do not seem to be adequately justified, e.g. ref 1 picked up on "Female mortality function differed from published ones (Fig. S2A)."

6) Ref 2 raised a related point on model validation

7) It is not possible to fully deduce what was done e.g. ref 1 mentioned it is that the authors did not fully specify what was done for the control measures: this is crucial to the main conclusions of the paper. I felt in my own reading that the reader is left to do a lot of inference to guess exactly how the model was implemented.

8) Justification for some modelling choices not adequately given, or alternatively the importance of these decisions mentioned (e.g. the 2% mentioned by ref 2)

9) Some key recent literature missed (2017 paper mentioned by ref 1)

Reviewer(s)' Comments to Author:

Referee: 1

Comments to the Author(s)

The authors of this paper develop and implement an age-structured model of tsetse fly population dynamics. They are interested in optimizing control strategies and hypothesize that failure to account for spatial variability could lead to population resurgence in unexpected areas. Failure to do area-wide interventions can lead to re-invasion by neighboring populations.

The paper is generally well written, but I think some of the writing could be revised for clarity. That is, there are some key terms which are not defined and this could make the paper hard to follow at points – e.g. cell, sources, sinks, patches, refuges. In addition, there are a number of phrases used which are unclear to me – e.g. “landscape configuration,” “rescue effect,” “grid scale.” Some sentences are unclear – see notes below.

My major concern is that the authors may have overstated some of their results. It seems like the authors had only one model with many constrained parameters. While some sensitivity analysis was performed (lines 125-6) and they do compare fits with observed age-structure, they do so only for intermediate aged flies and this model does not appear to be fit or calibrated to data; it was only stated that the population levels appeared to be close to the observed levels. While in a modeling context it's difficult to explore every possible scenario, since the authors didn't explore several model structures and/or parameter sets, it's unclear to me how sensitive their results are to the specific model assumptions and parameters they chose. I.e. two reasonably well-calibrated models can produce dramatically different results under intervention.

I'm particularly skeptical of the result that “female mortality function differed from published one” – field and experimental observations informed model inputs, so this seems somewhat inconsistent to favor certain observations over others. Furthermore, it's not clear to me based on this statement alone that field observations of mortality are flawed because a model required different mortalities.

Lastly, some minor concerns:

- The control strategies are not entirely clear to me. Were all combinations of cells tried at all levels? This seems like too many possible combinations to examine. I.e. which 70%/46% of the area was controlled? (Lines 187-8). How was this decided?
- I recommend any values should have confidence intervals – i.e. at the very least, the variability described in the sensitivity analysis should be propagated through to final estimates. E.g. percent variance explained & graphs (e.g. figure 2A).

Minor Notes

- Abstract: Spatio-*temporal* and dynamical model seems redundant since a dynamical model must concern changes over time.
- Abstract: Spatial complexity, or spatial variability? I would change this to variability.
- Abstract: what is a cell? I gather it's a pixel on your graph in figure X?
- Introduction, lines 5-6: “Confounding the role of spatial and temporal environmental heterogeneity could induce erroneous predictions of ecological processes. Unclear. Rephrase?”
- Line 9: Replace x. I.e. write out “vector and host interactions”
- What is “landscape configuration”? “Rescue effect”?
- Clear definitions of cells, sources, sinks, and patches would be helpful.
- Lines 18-19: Revise sentences – e.g. -> They are widely distributed *across* Africa*,* *occurring* in 38 countries *and* infesting around 10 million km².
- Lines 21-22: This sentence is unclear: “Among the 31 species and subspecies of tsetse flies, a third is of economic and human health importance”

- Line 23: on-going -> ongoing
- Lines 29-30: Not sure what scanned means here.
- decision-making -> decision making
- Line 72: pest management approaches -> pest-management approaches
- Line 73: spatio-temporal population dynamics seems redundant
- Line 104: Define cell
- The list of tsetse models doesn't seem exhaustive & I am skeptical that there are so few modeling papers (e.g. Hargrove JW has a similar tsetse modeling paper from 2017)
- Lines 152-3: What is the logic in using a different pupal emergence function
- Results line 169-70: Confidence intervals on variance explained?
- Discussion: "knowledge-driven" and "data-driven" – not sure the usage here makes sense.
- Line 249 – what is meant by "binary occupancy"

Referee: 2

Comments to the Author(s)

This is a very good paper on an interesting and impactful disease, with the interaction of temperature and environmental conditions on vector borne disease being a subject of broad general importance.

I only have one more substantial comment, and this is in regards to the model validation. The authors state that partial validation was done by comparison of model age structure to field data and later that this was good and robust as it did not vary with parameters (line 244). I do find this slightly concerning, as the lack of variation would suggest it is not 'important' in the model, in the sense that other factors of importance do vary with parameters (i.e. what has been validated may be a robust quantity, but if it doesn't vary with outputs of interest as the parameters vary, how much is it telling us about the critical parts of the model). Therefore it would be helpful to identify other points of validation, preferably ones where there is greater sensitivity to parameters. Alternatively if the authors can show why age structure is a key point of validation, then that would suffice.

A further smaller point is that the proportionate area of control was chosen for an entirely rational but seemingly arbitrary reason - why is 2% of the population important (I presume the figure is used because of historical levels of control as in line 24, but this does not seem to have epidemiological importance)? If there is a reason beyond this, it would be helpful to have this clarified. Also, it seems likely that, in poorer countries, in practice this level of control of 46% in large areas seems like it might be rather high - if this is not the case, it would be worthwhile to have a brief point made in the paper. Otherwise it would seem to be worthwhile to ask the question from the point of view of a more 'practical' coverage. An interesting question in this regard given the distribution of refuges around the heterogeneous control areas would be whether or not a buffer zone (essentially a band around the edges of the top 46% area) could usefully protect spillover areas even if the total control area was smaller, given the very short daily dispersal ranges.

a few minor points follow:

line 10 should be 'and thus'

line 27 should be 'and helps us to'

line 55 is it the decrease of the temperature over the period between larvipositions that matters, or is it mean here that the period is lower at lower temperatures?

line 58 should be capital 'C' for temperatures (2x)

line 61 'to their first host'

line 101 how was the grid cell size chosen? It looks like the cell size is well below the scale at which fly population effects typically vary (at least in the figure) but it would be helpful if this relationship were quantified, e.g by something like a measure of the variation of key quantities within a cell, compared to the variation between cells. if the former is high compared to the latter, it would suggest you might be missing some important effects

line 196 the lack of importance of temperature doesn't seem intuitive to me and is therefore interesting - it would be helpful to have a bit more discussion this.

Author's Response to Decision Letter for (RSPB-2019-1735.R0)

See Appendix A.

RSPB-2020-0749.R0

Review form: Reviewer 2

Recommendation

Accept with minor revision (please list in comments)

Scientific importance: Is the manuscript an original and important contribution to its field?

Good

General interest: Is the paper of sufficient general interest?

Good

Quality of the paper: Is the overall quality of the paper suitable?

Good

Is the length of the paper justified?

Yes

Should the paper be seen by a specialist statistical reviewer?

No

Do you have any concerns about statistical analyses in this paper? If so, please specify them explicitly in your report.

No

It is a condition of publication that authors make their supporting data, code and materials available - either as supplementary material or hosted in an external repository. Please rate, if applicable, the supporting data on the following criteria.

Is it accessible?

N/A

Is it clear?

N/A

Is it adequate?

N/A

Do you have any ethical concerns with this paper?

No

Comments to the Author

The paper is clearer and the previous comments addressed. I do have one additional concern that has arisen from the revised manuscript. As it stands, it is not clear as to the extent to which the spatial element of the model is important. The grid squares that the authors have used seem to simply divide the population into two areas, some viable, some not, with a boundary layer in between. This is not an issue in itself, however what it does not address is whether or not the persistence of vector populations is locally driven (that is, by local parameters) or whether the interaction between the cells is driving persistence. There are hints in the paper but it is not discussed explicitly and it would be useful to have this discussion. In particular if it is the local within-cell scale that is most important, then a spatial model is arguably not necessary at all (just a map of local conditions). However if between cell persistence is maintaining the population, then the need for the spatial model is clear.

A few minor points below:

In the abstract should be 'variations'

line 24 should be US\$

line 47 should be 'increases'

line 71 - why concentrate on gambiensis.

line 285 "variation"

line 88 onwards - some references supporting these statements would help especially for the broader audience of the journal.

line 99 - where are the trap catch data described? and would it also have seasonal dependence?

Review form: Reviewer 3 (John Hargrove)

Recommendation

Major revision is needed (please make suggestions in comments)

Scientific importance: Is the manuscript an original and important contribution to its field?

Acceptable

General interest: Is the paper of sufficient general interest?

Acceptable

Quality of the paper: Is the overall quality of the paper suitable?

Acceptable

Is the length of the paper justified?

Yes

Should the paper be seen by a specialist statistical reviewer?

No

Do you have any concerns about statistical analyses in this paper? If so, please specify them explicitly in your report.

Yes

It is a condition of publication that authors make their supporting data, code and materials available - either as supplementary material or hosted in an external repository. Please rate, if applicable, the supporting data on the following criteria.

Is it accessible?

No

Is it clear?

No

Is it adequate?

No

Do you have any ethical concerns with this paper?

No

Comments to the Author

See attached file below.

Review form: Reviewer 4 (Timothée Vergne)

Recommendation

Accept with minor revision (please list in comments)

Scientific importance: Is the manuscript an original and important contribution to its field?

Excellent

General interest: Is the paper of sufficient general interest?

Good

Quality of the paper: Is the overall quality of the paper suitable?

Good

Is the length of the paper justified?

Yes

Should the paper be seen by a specialist statistical reviewer?

No

Do you have any concerns about statistical analyses in this paper? If so, please specify them explicitly in your report.

No

It is a condition of publication that authors make their supporting data, code and materials available - either as supplementary material or hosted in an external repository. Please rate, if applicable, the supporting data on the following criteria.

Is it accessible?

Yes

Is it clear?

Yes

Is it adequate?

Yes

Do you have any ethical concerns with this paper?

No

Comments to the Author

This manuscript presents thoroughly a deterministic compartmental model of tsetse fly population dynamic that accounts for a spatially heterogeneous environment (through a regular grid of 900 cells), movement of flies between cells and a density dependence of the fly dynamic. It includes a sensitivity analysis of the model input parameters on population size that demonstrates that the population dynamic is mostly driven by temperature and adult mortality. Simulations were subsequently conducted to evaluate the impact of an increase of the adult mortality on the reduction of the population, assuming either a spatially homogeneous control strategy or a heterogeneous control strategy targeting the cells with the highest carrying capacity.

I was one of the reviewers of a previous version of this manuscript when it was submitted to Peers Community in Ecology, and have already made my major comments at that time. I found the current version very strong even if it can be sometimes difficult to follow between the main text and the supplementary materials. But I am aware of the length restrictions, so I think it is fine as it is.

I only have minor comments to be addressed before the manuscript can be accepted for publication:

Abstract: From "The coldest cells" to "related to temperature" is difficult to follow and to understand as the sentences are too long. Consider rephrasing them.

L87: I assume "that" refers to "population age structure" and not to "abortion rates". Please clarify.

L120-121: Suggestion "We used a constant rate to model pupa mortality, given the lack of data on this parameter [31]". Otherwise it is not clear what this constant rate is referring to.

Figure 1: I , nP , nN and nF should be defined in the legend. Also consider including " $S \in \{P, N, Fx, M\}$ " when you first mention "for stage S ".

L156: "to 5% and 2%"

L170-178: you present Figures S1A and S2B but not Figure S2C. Talking about S2C, given the amount of observed data that were used to calibrate the model of the time to larviposition and the fact that they all relate to 25°C, the relationship with the temperature seems very far-fetched to me. It seems that there is no evidence to justify this association. You should probably consider the time to larviposition as a constant, equal to 18-19 days for nulliparous and 10 days for parous females. Just like you did for pupa mortality.

L210: "smaller increase in mortality"

LL323-327: I suggest you include a discussion on the feasibility of implementing heterogeneous control (as opposed to homogeneous)

SuppMat 2.3: please include the corresponding reference.

Decision letter (RSPB-2020-0749.R0)

20-Jul-2020

Dear Dr Ezanno,

Thank you for the revised version of this manuscript, and my apologies for the delay in returning a decision to you on it.

We received conflicting reviewers' opinions on the revised version, and, as the Associate Editor who had dealt with the first version is now fully engaged with Covid19 responses, I needed to ask the advice of a new AE. The new AE shares many of the reservations of the more critical referee on this version. I full appreciate that you have put a lot of work into addressing the comments on the first version, and it is always regrettable to have to ask for further substantial changes in a second round of revisions. But I'm afraid that several major issues have been raised at this stage, so I am not able to accept the manuscript at present: major revisions will be needed, which I want to be sure that you have the time to complete.

The manuscript has therefore been rejected for publication in Proceedings B in its current stage, but we invite you to submit a revised version, provided the comments of the referees are fully addressed. However please note that this is not a provisional acceptance.

You will see that the new (critical) reviewer has provided an exceptionally detailed review, and some important positive notes of encouragement at the end (whilst also excusing themselves as an 'elderly curmudgeon!'); I hope all aspects will help with the revision.

The resubmission will be treated as a new manuscript. However, we will approach the same reviewers if they are available and it is deemed appropriate to do so. Please note that resubmissions must be submitted within six months of the date of this email. In exceptional circumstances, extensions may be possible if agreed with the Editorial Office.

Please find below the comments made by the referees, not including confidential reports to the Editor, which I hope you will find useful. When you resubmit your manuscript, please upload the following:

- 1) A 'response to referees' document including details of how you have responded to the comments, and the adjustments you have made.
- 2) A clean copy of the manuscript and one with 'tracked changes' indicating your 'response to referees' comments document.
- 3) Line numbers in your main document.
- 4) Please read our data sharing policies to ensure that you meet our requirements <https://royalsociety.org/journals/authors/author-guidelines/#data>.

My apologies again about the length of this process, which I'm afraid is largely a reflection of the difficult global situation. I hope you and all your co-authors are well, and we will look forward to receiving a revised version of this paper.

Yours sincerely,
Professor Loeske Kruuk
Editor

Associate Editor

Comments to Author:

I have now received three reviews including a new one that raises substantial issues concerning the parameterization, assumptions and interpretation of your results. Everyone agrees that the model is very interesting and is a good contribution to the field. You have also done a good job clearing up issues raised by previous reviewers. However, my sense is that the detailed critical review has clearly raised a number of important points that need to be addressed. In my reading of the paper I also found the scattered statements on the effectiveness of SIT somewhat out of place in the manuscript. You are not really addressing the relative effectiveness of control strategies in any direct way and it was unclear to me why these assertions were in the paper. It may be that you need to directly address this with specific modeling.

Many of the other points by the critical reviewer may well be addressed with either rewrites and some modeling. One of the previous reviewers also makes an important point about the need for the spatial model and therefore whether spatial structure is playing a role here. I was unsure what the relative role of space is in the model – clearly there is a big effect of environment and it is realistic to have this varying in space, but is it the case that it is environmental heterogeneity rather than space that matters? How many of your results could be obtained with just the environmental heterogeneity that you show? I think you could give the readers a good sense of that. Both of these referees make some minor comments that would improve the manuscript.

In terms of the papers' suitability for PRSB, you need to address the issues of the critical reviewer, and also be clear why the paper is more than a specific model showing how environmental factors may impact the control of a particular pest. The paper may make a general impact in pushing the approach of using data driven environmental parameterization of vector control models, but it may have a bigger impact in a more focused journal. You need to be clear about the general insights from the model beyond the system.

Reviewer(s)' Comments to Author:

Referee: 2

Comments to the Author(s).

The paper is clearer and the previous comments addressed. I do have one additional concern that has arisen from the revised manuscript. As it stands, it is not clear as to the extent to which the spatial element of the model is important. The grid squares that the authors have used seem to simply divide the population into two areas, some viable, some not, with a boundary layer in between. This is not an issue in itself, however what it does not address is whether or not the persistence of vector populations is locally driven (that is, by local parameters) or whether the interaction between the cells is driving persistence. There are hints in the paper but it is not discussed explicitly and it would be useful to have this discussion. In particular if it is the local within-cell scale that is most important, then a spatial model is arguably not necessary at all (just a map of local conditions). However if between cell persistence is maintaining the population, then the need for the spatial model is clear.

A few minor points below:

In the abstract should be 'variations'

line 24 should be US\$

line 47 should be 'increases'

line 71 - why concentrate on gambiensis.

line 285 "variation"

line 88 onwards - some references supporting these statements would help especially for the broader audience of the journal.

line 99 - where are the trap catch data described? and would it also have seasonal dependence?

Referee: 3

Comments to the Author(s).
See attached file below.

Referee: 4

Comments to the Author(s).

This manuscript presents thoroughly a deterministic compartmental model of tsetse fly population dynamic that accounts for a spatially heterogeneous environment (through a regular grid of 900 cells), movement of flies between cells and a density dependence of the fly dynamic. It includes a sensitivity analysis of the model input parameters on population size that demonstrates that the population dynamic is mostly driven by temperature and adult mortality. Simulations were subsequently conducted to evaluate the impact of an increase of the adult mortality on the reduction of the population, assuming either a spatially homogeneous control strategy or a heterogeneous control strategy targeting the cells with the highest carrying capacity.

I was one of the reviewers of a previous version of this manuscript when it was submitted to Peers Community in Ecology, and have already made my major comments at that time. I found the current version very strong even if it can be sometimes difficult to follow between the main text and the supplementary materials. But I am aware of the length restrictions, so I think it is fine as it is.

I only have minor comments to be addressed before the manuscript can be accepted for publication:

Abstract: From "The coldest cells" to "related to temperature" is difficult to follow and to understand as the sentences are too long. Consider rephrasing them.

L87: I assume "that" refers to "population age structure" and not to "abortion rates". Please clarify.

L120-121: Suggestion "We used a constant rate to model pupa mortality, given the lack of data on this parameter [31]". Otherwise it is not clear what this constant rate is referring to.

Figure 1: l , nP , nN and nF should be defined in the legend. Also consider including " $S \in \{P, N, Fx, M\}$ " when you first mention "for stage S ".

L156: "to 5% and 2%"

L170-178: you present Figures S1A and S2B but not Figure S2C. Talking about S2C, given the amount of observed data that were used to calibrate the model of the time to larviposition and the fact that they all relate to 25°C, the relationship with the temperature seems very far-fetched to me. It seems that there is no evidence to justify this association. You should probably consider the time to larviposition as a constant, equal to 18-19 days for nulliparous and 10 days for parous females. Just like you did for pupa mortality.

L210: "smaller increase in mortality"

LL323-327: I suggest you include a discussion on the feasibility of implementing heterogeneous control (as opposed to homogeneous)

SuppMat 2.3: please include the corresponding reference.

Author's Response to Decision Letter for (RSPB-2020-0749.R0)

See Appendix B.

RSPB-2020-2810.R0

Review form: Reviewer 2

Recommendation

Accept as is

Scientific importance: Is the manuscript an original and important contribution to its field?

Good

General interest: Is the paper of sufficient general interest?

Good

Quality of the paper: Is the overall quality of the paper suitable?

Good

Is the length of the paper justified?

Yes

Should the paper be seen by a specialist statistical reviewer?

No

Do you have any concerns about statistical analyses in this paper? If so, please specify them explicitly in your report.

No

It is a condition of publication that authors make their supporting data, code and materials available - either as supplementary material or hosted in an external repository. Please rate, if applicable, the supporting data on the following criteria.

Is it accessible?

N/A

Is it clear?

N/A

Is it adequate?

N/A

Do you have any ethical concerns with this paper?

No

Comments to the Author

Thank you for your extensive revisions. The spatial results are indeed interesting and well worth highlighting.

Review form: Reviewer 4

Recommendation

Accept with minor revision (please list in comments)

Scientific importance: Is the manuscript an original and important contribution to its field?

Excellent

General interest: Is the paper of sufficient general interest?

Good

Quality of the paper: Is the overall quality of the paper suitable?

Excellent

Is the length of the paper justified?

Yes

Should the paper be seen by a specialist statistical reviewer?

No

Do you have any concerns about statistical analyses in this paper? If so, please specify them explicitly in your report.

No

It is a condition of publication that authors make their supporting data, code and materials available - either as supplementary material or hosted in an external repository. Please rate, if applicable, the supporting data on the following criteria.

Is it accessible?

Yes

Is it clear?

Yes

Is it adequate?

Yes

Do you have any ethical concerns with this paper?

No

Comments to the Author

I was one of the reviewers of versions of this manuscript submitted to Peers Community in Ecology and RSPB, and have already made my major comments at that time. I found the current version very strong even if it can be sometimes difficult to follow between the main text and the supplementary materials. But I am aware of the length restrictions, so I think it is fine as it is. Congratulations on the work. I am looking forward to seeing this published after all your efforts. Three general comments hereafter.

1. Despite the length restriction, it would be paramount to state in the abstract what characterised the targeted locations for heterogeneous control and to clarify policy take-home message for tsetse control in a heterogeneous landscape.

2. My understanding is that mortality was increased and kept constant for the whole year of intervention, rather than implemented as discrete-time intervention waves, as one would expect interventions to be implemented. Could you justify your approach and add a couple of lines in

the discussion about how relevant this approach is with regards to actual on-site intervention strategies?

3. It is not fully clear of the different intervention strategies are comparable in terms of “effort” deployed. Indeed, I would assume that increasing mortality by 1.7 in 50% of cells does not require the same effort than increasing mortality by 1.6 in 100% of cells. Is there an objective way of measuring the “cost”? You could add this in your discussion for perspectives, if relevant.

Decision letter (RSPB-2020-2810.R0)

04-Jan-2021

Dear Dr Ezanno

I am pleased to inform you that your manuscript RSPB-2020-2810 entitled "Dispersal in heterogeneous environments drives population dynamics and control of tsetse flies" has been accepted for publication in Proceedings B. My apologies in the delay in sending you this outcome, which was largely due to me being out of reception on fieldwork and then Christmas leave.

The referees have recommended publication, but have also suggested some minor revisions to your manuscript. Therefore, I invite you to respond to the referees' comments and revise your manuscript. Because the schedule for publication is very tight, it is a condition of publication that you submit the revised version of your manuscript within 7 days. If you do not think you will be able to meet this date please let us know.

When submitting your revised manuscript, you will be able to respond to the comments made by the referee(s) and upload a file "Response to Referees". You can use this to document any changes you make to the original manuscript. We require a copy of the manuscript with revisions made since the previous version marked as ‘tracked changes’ to be included in the ‘response to referees’ document.

- 1) A text file of the manuscript (doc, txt, rtf or tex), including the references, tables (including captions) and figure captions. Please remove any tracked changes from the text before submission. PDF files are not an accepted format for the "Main Document".
- 2) A separate electronic file of each figure (tiff, EPS or print-quality PDF preferred). The format should be produced directly from original creation package, or original software format. PowerPoint files are not accepted.
- 3) Electronic supplementary material: this should be contained in a separate file and where possible, all ESM should be combined into a single file. All supplementary materials accompanying an accepted article will be treated as in their final form. They will be published alongside the paper on the journal website and posted on the online figshare repository. Files on

figshare will be made available approximately one week before the accompanying article so that the supplementary material can be attributed a unique DOI.

Thank you for submitting your manuscript to Proceedings B and I look forward to receiving your revision. If you have any questions at all, please do not hesitate to get in touch.

Finally, all the best for 2021, in the hope that it will be a calmer year than 2020,

Yours sincerely,

Professor Loeske Kruuk

Editor

Associate Editor

Board Member

Comments to Author:

Thank you for your detailed replies to the referees and the excellent revision of the paper. I think it is much improved and you have made positive changes in response to the critical referee where you can. The two referees who have looked at this version are positive. There are a few relatively minor comments from one of the referees for you to consider that will further improve the manuscript. I enjoyed reading this version of the paper.

Reviewer(s)' Comments to Author:

Referee: 2

Comments to the Author(s).

Thank you for your extensive revisions. The spatial results are indeed interesting and well worth highlighting.

Referee: 4

Comments to the Author(s).

I was one of the reviewers of versions of this manuscript submitted to Peers Community in Ecology and RSPB, and have already made my major comments at that time. I found the current version very strong even if it can be sometimes difficult to follow between the main text and the supplementary materials. But I am aware of the length restrictions, so I think it is fine as it is. Congratulations on the work. I am looking forward to seeing this published after all your efforts. Three general comments hereafter.

1. Despite the length restriction, it would be paramount to state in the abstract what characterised the targeted locations for heterogeneous control and to clarify policy take-home message for tsetse control in a heterogeneous landscape.
2. My understanding is that mortality was increased and kept constant for the whole year of intervention, rather than implemented as discrete-time intervention waves, as one would expect interventions to be implemented. Could you justify your approach and add a couple of lines in the discussion about how relevant this approach is with regards to actual on-site intervention strategies?
3. It is not fully clear of the different intervention strategies are comparable in terms of "effort" deployed. Indeed, I would assume that increasing mortality by 1.7 in 50% of cells does not require the same effort than increasing mortality by 1.6 in 100% of cells. Is there an objective way of measuring the "cost"? You could add this in your discussion for perspectives, if relevant.

Author's Response to Decision Letter for (RSPB-2020-2810.R0)

See Appendix C.

Decision letter (RSPB-2020-2810.R1)

08-Jan-2021

Dear Dr Ezanno

I am pleased to inform you that your manuscript entitled "Dispersal in heterogeneous environments drives population dynamics and control of tsetse flies" has been accepted for publication in Proceedings B.

Open Access

Paper charges

Sincerely,

Proceedings B

Appendix A

Proceedings B - Manuscript ID RSPB-2019-1735 (resubmission)

First of all, we would like to thank the two referees for their helpful comments on our manuscript. We have addressed them in the new version of the manuscript. Please find thereafter in blue our response to these comments as well as the editor's ones, also indicating where the manuscript has been modified (line numbers).

Associate Editor

Board Member: 1

Combining the comments from the referees and my own reading: this paper is on an interesting and important topic, but there are a number of major issues with the manuscript. In summary of these:

1) There are issues with clarity of the paper, in particular terms not defined clearly enough, ref 1 refers to some. 2) In many places the writing is not easy to decipher, e.g. sentences in L61-62:

The paper has been entirely read and corrected by a native English speaker (M. Vreysen, co-author of the manuscript). Terms that were unclear have been changed or defined.

Learning capability of older flies makes them return on their first host, increasing their hunting efficiency. (Why this example is not clear: Maybe there's a grammar error ("on" rather than "in"), but also what is "learning capability". And specifically returning to the first host, presumably to feed? And no explanation of why that increases their hunting efficiency.)

Especially this sentence has been changed to (L66-68): "Acquired feeding preferences have been demonstrated in this species, i.e., the host selected for the first blood meal can influence the one selected for the second meal. This learning capability increases the hunting efficiency of older flies [52]."

3) Too much key information is left to the SI. In particular, it is not possible to deduce from the main text the general form of the model used, and this is important given it is the heart of what was done.

To help the reader understand more easily the form of the model, we have added in the main text (new Fig. 1) the model figure that was previously in the SI (previous Fig. S5). To keep the number of figures convenient for PRSB, we have moved the figure on the sensitivity analysis (previous Fig. 1) to SI (new Fig. S9). We made the model as clear as possible in the main text. However, it is a rather complex model, which description is long. Due to size limitation for the main text, we decided to focus on key model assumptions and results, as readers of PRSB are not all modellers. However, the model is fully described in the SI (SI 3-6). We hope this will be convenient.

4) Ref 1 raised that the author's results may be overstated: specifically sensitivity analysis was limited, and no check of sensitivity to model choice. I would agree with this. See Ref 1's comments for details.

The model is a mechanistic model. This means that its structure is based on all of the knowledge available to date on the species modelled, using processes that all have a biological meaning. Therefore, there is not several possible model structures among which we could hesitate here. For processes modelled by a function (e.g. temperature-dependent), the function parameters were all calibrated on observed data as described in the SI. It is very rare for this kind of model to analyse its sensitivity to variations in its structure, most of the effort generally focusing on variations in the parameter values, as we did. With regards to model sensitivity to parameter variations, we chose the most relevant type of analysis, i.e. a variance-based global sensitivity analysis. In this kind of analysis, all the parameters vary simultaneously (here we used a FAST sampling design). This enables us to calculate the principal sensitivity indices (one per parameter), as well as the total sensitivity indices (main effect of the parameter plus interactions in which it is involved).

5) Related to 4) and maybe 2)... In many places there were remarks that do not seem to be adequately justified, e.g. ref 1 picked up on "Female mortality function differed from published ones (Fig. S2A)."

We justified more the choices made (see answer to referees' comments).

6) Ref 2 raised a related point on model validation

This was also addressed (see answer to referees' comments).

7) It is not possible to fully deduce what was done e.g. ref 1 mentioned it is that the authors did not fully specify what was done for the control measures: this is crucial to the main conclusions of the paper. I felt in my own reading that the reader is left to do a lot of inference to guess exactly how the model was implemented.
8) Justification for some modelling choices not adequately given, or alternatively the importance of these decisions mentioned (e.g. the 2% mentioned by ref 2)

For points 7 & 8: what was done is fully described in the SI (section 6). Due to size limitation of the main text, we focused there on key aspects and results obtained, while the exact methods are described in SI to make the study reproducible. We revised SI and the main text to make it clearer.

9) Some key recent literature missed (2017 paper mentioned by ref 1)

Please, see thereafter the detailed answer to ref 1.

Referee: 1

The authors of this paper develop and implement an age-structured model of tsetse fly population dynamics. They are interested in optimizing control strategies and hypothesize that failure to account for spatial variability could lead to population resurgence in unexpected areas. Failure to do area-wide interventions can lead to re-invasion by neighboring populations.

Thank you for your detailed comments. We hope to have addressed them in a satisfactory way, so that the paper becomes clearer.

The paper is generally well written, but I think some of the writing could be revised for clarity. That is, there are some key terms which are not defined and this could make the paper hard to follow at points—e.g. cell, sources, sinks, patches, refuges. In addition, there are a number of phrases used which are unclear to me—e.g. “landscape configuration,” “rescue effect,” “grid scale.” Some sentences are unclear—see notes below.

The paper has been entirely read and corrected by a native English speaker (M. Vreysen, co-author of this manuscript). Terms that were unclear have been changed or defined (see precisions thereafter). Nevertheless, most of the readership of PRSB is highly skilled in ecology and most of these terms are from this field thus should not be too problematic here. We used “cell” throughout the method section (and SI) which corresponds to the spatial unit in the model to distinguish from “patch” which can be of any size (e.g. a cell or also a group of cells). We did not use “pixels” as this term was seen as jargon by colleagues from ecology field.

My major concern is that the authors may have overstated some of their results. It seems like the authors had only one model with many constrained parameters. While some sensitivity analysis was performed (lines 125-6) and they do compare fits with observed age-structure, they do so only for intermediate aged flies and this model does not appear to be fit or calibrated to data; it was only stated that the population levels appeared to be close to the observed levels. While in a modeling context it’s difficult to explore every possible scenario, since the authors didn’t explore several model structures and/or parameter sets, it’s unclear to me how sensitive their results are to the specific model assumptions and parameters they chose. I.e. two reasonably well-calibrated models can produce dramatically different results under intervention.

We did not perform parameter inference by minimizing the difference between model outputs and validation data because such data were lacking. Instead, we calibrated on data each of the equations explicitly representing the processes involved in the population dynamics, namely adult mortality, pupal development, and time to larvipositions. For this, we used the most relevant data available on our species of interest, some being published, some being new. This was not sufficiently clearly stated. The legend of Fig. S2 thus has been revised as well as the main text (L109-111). As a result, only one reasonably-well calibrated model can be obtained, which integrates all of the relevant processes known so far related to the population dynamics of this species. Hence, no alternative model structure could have been tested (now said L142). However, it is true that some parameters are harder to calibrate than others. This is why we performed a sensitivity analysis, testing several parameter sets. We followed a global design as described in Saltelli et al. (2008), fully described in Supplementary Information 5, which was recommended for this type of model [65]. Parameters varied simultaneously within a $\pm 5\%$ range (except for temperature $\pm 0.3^\circ\text{C}$). We then assessed how these variations impacted model outputs of interest.

Concerning the comparison to the observed age structure, the limitation to intermediate-age flies is justified L147: “Predicted age structure was compared with field data for females of ovarian age 1, 2, and 3, because

traps do not catch nulliparous and females of ovarian age 4 and more as efficiently as females of intermediate ovarian ages [66].”

I’m particularly skeptical of the result that “female mortality function differed from published one”—field and experimental observations informed model inputs, so this seems somewhat inconsistent to favor certain observations over others. Furthermore, it’s not clear to me based on this statement alone that field observations of mortality are flawed because a model required different mortalities.

We used all data at our disposal on our species of interest, either already published or new, to calibrate the equation of the female temperature-dependent mortality. Data were obtained on a temperature range from 24 to 32°C (in Figure S2A, in purple for new data, in grey for published data). Only one outlier was removed. The new equation obtained using these data (purple line) differed from published ones (blue line, light and dark orange lines). The blue line corresponds to an equation used in Barclay and Vreysen 2011, 2013, based on expert’s opinion, with no data provided. In addition, this equation is said to be adapted to both *G. palpalis* spp. and *G. pallidipes*, while differing from the equation tailored to *G. pallidipes* (dark orange line) published in Hargrove 2004 and based on data from mark-recapture experiments. Finally, the data we used to calibrate this equation were obtained from lab experiments directly measuring mortality, which is reliable and less subject to bias induced by inference made on field observations, as discussed in Hargrove and Ackley 2015. We modified Fig. S2 and its legend to make it clearer, and stated it more clearly in the main text (L170).

Lastly, some minor concerns:

- The control strategies are not entirely clear to me. Were all combinations of cells tried at all levels? This seems like too many possible combinations to examine. I.e. which 70%/46% of the area was controlled? (Lines 187-8). How was this decided?

We have not tested all combinations of treated cells for all proportions of treated space. To minimize the number of scenarios to be tested, the proportion of treated cells gradually decreased while mortality gradually increased. For a given proportion (x%), the cells to be treated were selected from those also treated for the scenario with the next highest proportion tested (y%>x%). The cells removed from treatment were preferentially those where the treatment had been least effective in scenario “y%”. Mortality was then gradually increased until the female population was reduced to 2 or 5% of its original size. If this level of reduction was not achievable at a given proportion of cells to be treated, then we did not test for even lower proportions.

The sentence was rephrased (now L211-213): “Controlling 70% of well-chosen cells was as efficient as controlling the entire area (almost the same mortality was applied).”

The SI (section 6) fully described how control was implemented and how scenarios were defined. We added the explanation provided here before to make it clearer.

- I recommend any values should have confidence intervals—i.e. at the very least, the variability described in the sensitivity analysis should be propagated through to final estimates. E.g. percent variance explained & graphs (e.g. figure 2A).

The sensitivity analysis concerned the reference scenario, not all the scenarios explored in the control section, which would have resulted in far too many numerical analyses. As the model is deterministic, the global variance-based sensitivity analysis conducted provide point estimate of the sensitivity indices, not confidence intervals. However, to give an idea of the variation of model outputs studied in this sensitivity analysis, their distributions are provided in the SI (section 5).

Minor Notes

- Abstract: Spatio-***temporal*** and dynamical model seems redundant since a dynamical model must concern changes over time.

Changed to: “a mechanistic spatio-temporal model of the age-structured population dynamics of tsetse flies”

- Abstract: Spatial complexity, or spatial variability? I would change this to variability.

Changed

- Abstract: what is a cell? I gather it’s a pixel on your graph in figure X?

A cell is the spatial unit in our model. Thus yes it corresponds to a pixel in the map. However, we preferred using a single term throughout the paper to make it clearer and chose “cell”. In the abstract, we define this term as follow: “integrating spatial variability among cells (the model spatial unit)”

- Introduction, lines 5-6: “Confounding the role of spatial and temporal environmental heterogeneity could induce erroneous predictions of ecological processes. Unclear. Rephrase?”

Rephrased to (L7-8): “Spatial and temporal environmental heterogeneities have distinct effects. If they are confounded, this can lead to erroneous predictions of ecological processes [6].”

- Line 9: Replace x. I.e. write out “vector and host interactions”

Done

- What is “landscape configuration”? “Rescue effect”?

Landscape configuration has been changed to: “the structure of the landscape”.

Rescue effect is a well-known ecological process, thus the expression has been kept (and defined): “(i.e. when the arrival of new individuals rescues a local population from extinction)”.

- Clear definitions of cells, sources, sinks, and patches would be helpful.

“Cell” has been defined in the abstract and as well at first use in main text.

“Source patches”: (i.e. area where the local population increases)

“Sink patches”: (i.e. area where the local population decreases)

- Lines 18-19: Revise sentences—e.g. -> They are widely distributed ***across*** Africa*,* ***occurring*** in 38 countries ***and*** infesting around 10 million km².

Done

- Lines 21-22: This sentence is unclear: “Among the 31 species and subspecies of tsetse flies, a third is of economic and human health importance”

We have extended this sentence to make it clearer: “..., the remaining mostly thriving in wildlife areas or located in thinly populated forested areas with little or no livestock”.

- Line 23: on-going -> ongoing

Done

- Lines 29-30: Not sure what scanned means here.

Replaced by “scrutinized”.

- decision-making -> decision making

Done

- Line 72: pest management approaches -> pest-management approaches

Done

- Line 73: spatio-temporal population dynamics seems redundant

Indeed, the dynamics can be spatial, temporal or both, this is why we make it explicit here.

- Line 104: Define cell

Done

- The list of tsetse models doesn't seem exhaustive & I am skeptical that there are so few modeling papers (e.g. Hargrove JW has a similar tsetse modeling paper from 2017)

We did not aim at making a comprehensive review of modelling papers on tsetse populations dynamics, but rather at focusing on the different kinds of models proposed so far, and especially those with spatial dynamics. However, we identified three missing papers which we added in the discussion where they were the most useful. Thereafter, we provide more details on why some papers were or not cited in our own paper.

Several modelling papers can use the same model (same structure and assumptions), e.g. the model in Vale and Torr 2005 (diffusion in 1D, cited in our paper) is also used in Torr and Vale 2011 (Is the even distribution of insecticide-treated cattle essential for tsetse control? Modelling the impact of baits in heterogeneous environments. Plos Neglected Tropical Diseases). Furthermore, models focusing on the epidemiology of Human African Trypanosomiasis generally provide a less precise parameterisation of tsetse life cycle. We did not cite associated modelling papers considered as too far from the scope of our paper:

- Hargrove et al. (2012) Modeling the Control of Trypanosomiasis Using Trypanocides or Insecticide-Treated Livestock. PLoS Negl Trop Dis 6(5): e1615.
- Alderton et al. (2016) A Multi-Host Agent-Based Model for a Zoonotic, Vector-Borne Disease. A Case Study on Trypanosomiasis in Eastern Province, Zambia. PloS Negl Trop Dis 10(12):e0005252
- Alderton S et al. (2018) An agent-based model of tsetse fly response to seasonal climatic drivers: Assessing the impact on sleeping sickness transmission rates. PloS Negl Trop Dis 12(2):e0006188. This paper uses Alderton et al. 2016 model, which is cited in our paper.
- Meisner et al. (2019) A mathematical model for evaluating the role of trypanocide treatment of cattle in the epidemiology and control of *Trypanosoma brucei rhodesiense* and *T. b. gambiense* sleeping sickness in Uganda. Parasite Epidemiology and Control, e00106.
- Castaño et al. (2020) Assessing the impact of aggregating disease stage data in model predictions of human African trypanosomiasis transmission and control activities in Bandundu province (DRC). PloS Negl Trop Dis 14(1):e0007976

We added the following papers:

- Ackley and Hargrove 2017 (A dynamic model for estimating adult female mortality from ovarian dissection data for the tsetse fly *Glossina pallidipes Austen* sampled in Zimbabwe): it is a mechanistic model used to infer parameter values from field data. This model does not account for temperature-dependence, larviposition times are assumed constant, and the model does not account for the spatial dynamics of the population.
- Lord et al. 2017 (Climate change and African trypanosomiasis vector populations in Zimbabwe's Zambezi Valley: A mathematical modelling study): this model has the same structure as our model, but is calibrated to another species in another area. It also emphasizes the strong effect of the temperature, which increases in their study area and induces a decline in the population. However, this model does not represent the spatial dynamics of the population in a fragmented landscape as we did.
- Hargrove and Vale 2019 (Models for the rates of pupal development, fat consumption and mortality in tsetse (*Glossina* spp)): it is a new equation system derived from a reanalysis of data on *G. m. morsitans*, and consistent with some choices we made such as the sigmoidal form for pupae development. However, no mechanistic predictions were implemented using these equations, which can be seen as guidelines to parameterise future models. Authors also raise concerns about the difficulty to estimate temperatures actually experienced by tsetse flies, as we did, and comment on the paucity of information on tsetse species other than *G. m. morsitans*. We think our model nicely contributes to fulfil this lack of knowledge.

We modified the discussion accordingly (with numbers for reference here highlighted with authors' names to make it clearer): "The effect of temperature on fly population dynamics both at larger and local scales emphasises the need for further investigating the impact of climate change on tsetse populations [68-69]." (L261-2) and then: "Using a landscape predicted from remote sensing data and realistic temperature data, the importance of refuges was evidenced, which was previously not obvious. Published models barely account for the spatial dynamics of tsetse populations [31, 70], or only in one dimension while assuming theoretical carrying capacities [39]. A single spatial agent-based model has been proposed previously [29], which dynamically assigns patch suitability in a binary way (1 or 0) using environmental data, but which does not account for temperature-dependence in biological processes and only tests homogeneous control scenarios. In addition, recent field and laboratory data on mortality, development, and dispersal were incorporated to our model, decreasing the paucity of information on tsetse species other than *G. m. morsitans* [71]. Predicted age structure was in good agreement with field data, and proved robust as it barely varied following parameter variations. Amplitude and duration of seasons are expected to be major drivers influencing of ovarian age distribution of the population, but this could not be assessed here as temperature data were only available for one year. Our results highlight the need for more biological studies to better infer mortality variations with

temperature, and more accurately estimate temperatures as perceived by insects, an issue recently raised for another tsetse species [71]. Such a complementary interplay between models, field observations, and laboratory experiments is fundamental to make accurate predictions.” (L272-288)

- Lines 152-3: What is the logic in using a different pupal emergence function

The available emergence function (Hargrove 2004; Hargrove and Vale 2019) was provided for another species (*G. m. morsitans*). Thus, we used the data we had to fit a new function dedicated to *G. palpalis gambiensis*. A sigmoid function is much more realistic than an exponential one, and also relevant with regards to data, thus was retained here.

- Results line 169-70: Confidence intervals on variance explained?

See our answer to your related general comment.

- Discussion: “knowledge-driven” and “data-driven” —not sure the usage here makes sense.

This paragraph has been revised, see previous comment on modelling papers.

- Line 249—what is meant by “binary occupancy”

This paragraph has been revised, see previous comment on modelling papers.

Referee: 2

This is a very good paper on an interesting and impactful disease, with the interaction of temperature and environmental conditions on vector borne disease being a subject of broad general importance.

Thanks! We hope to have addressed your comments in a satisfactory way, so that the paper becomes clearer.

I only have one more substantial comment, and this is in regards to the model validation. The authors state that partial validation was done by comparison of model age structure to field data and later that this was good and robust as it did not vary with parameters (line 244). I do find this slightly concerning, as the lack of variation would suggest it is not 'important' in the model, in the sense that other factors of importance do vary with parameters (i.e. what has been validated may be a robust quantity, but if it doesn't vary with outputs of interest as the parameters vary, how much is it telling us about the critical parts of the model). Therefore it would be helpful to identify other points of validation, preferably ones where there is greater sensitivity to parameters. Alternatively if the authors can show why age structure is a key point of validation, then that would suffice.

We used two criteria to evaluate the reference scenario: (1) this scenario without control mimics a zone where the tsetse population is installed, thus the model should predict a “steady” population dynamics over the years (i.e., showing similar patterns among years), which is what has been obtained (Fig. S6B); (2) the comparison of predicted and observed age structures. While it was recently shown that ovarian age from field captured individuals should not be used to estimate mortality rates (Hargrove and Ackley 2015), the age structure remains an important ecological feature of a population, routinely measured in the field. Variations between model predictions and observations would thus have made us reject model parameterisation. The fact that the summary outputs related to the age structure did not vary in our sensitivity analysis does not enable us to refine parameter values (we cannot reject a lack of precision of $\pm 5\%$), but the good agreement indicates that the model parameterisation is acceptable. Indeed, the summary outputs vary when testing a wider range of parameter variation, e.g., when decreasing the development of nulliparous by 50% (i.e., increasing the time to produce the first offspring) (Fig. L1).

Figure L1: Predicted age structure (per day) in the reference scenario (left) and when decreasing the development of nulliparous by 50% (right). The impact of parameter variation is the highest at the beginning of the year, when temperatures are the lowest (see Fig. S6A), and thus induce constant mortalities and longer larviposition times.

It remains that the predicted age structure is highly robust to parameter variations especially for high temperatures. We agree that more data are required to better assess the model behaviour, which are lacking so far. This highlights also the need for a good estimation of temperatures as they highly impact predictions.

L88: we replaced “partial validation” by “qualitative evaluation” to avoid any confusion.

A further smaller point is that the proportionate area of control was chosen for an entirely rational but seemingly arbitrary reason - why is 2% of the population important (I presume the figure is used because of historical levels of control as in line 24, but this does not seem to have epidemiological importance)? If there is a reason beyond this, it would be helpful to have this clarified. Also, it seems likely that, in poorer countries, in practice this level of control of 46% in large areas seems like it might be rather high - if this is not the case, it would be worthwhile to have a brief point made in the paper. Otherwise it would seem to be worthwhile to ask the question from the point of view of a more 'practical' coverage. An interesting question in this regard given the distribution of refuges around the heterogeneous control areas would be whether or not a buffer zone (essentially a band around the edges of the top 46% area) could usefully protect spillover areas even if the total control area was smaller, given the very short daily dispersal ranges.

Vreysen et al. [14] explain the necessity to reduce densities to a low level (suppression phase) before turning to the sterile insect technique (SIT) for the final eradication component: “the SIT relies on [...] the sustained and systematic release of the sterile males over the target area in numbers large enough in relation to the wild male population to out-compete them for wild females”. The particular case of the Niayes in Senegal is also described, with the chosen techniques applied for the suppression phase: insecticide-based methods and trappings, coherent with the simulation of an increased mortality in our paper.

Dicko et al. [40] give the reduction of densities reached during the eradication campaign in the Niayes. After the suppression phase (what we intended to simulate) and before the eradication phase (consisting of SIT), the two studied « blocks » had achieved a 99,6 % and 90,4 % reduction, respectively. Kagbadouno et al. [76] also describe the eradication of tsetse (*Glossina palpalis gambiensis*, the same sub-species as in our paper) in the Loos islands, Guinea, with density reductions of 97 % and 98 % in 6 months observed in Room and Kassa islands, respectively. These levels are consistent with what we tested in our model (95 % and 98 % reduction). Concerning the 2 % mentioned L24, beware that it does not refer to the same aspect of control (the reduction level of the suppression phase). Indeed, it refers to the limited spatial extent of current successes of eradication campaigns. Allsopp 2001 [17] explains that “positive efforts to reduce tsetse distributions are currently being pursued in less than 2 % of the infested areas”. In Bouyer et al. [18], a table giving the impact of past tsetse control campaigns shows that most failed to reach eradication.

We extended a bit the text in the discussion to make this clearer (L304, L312). Due to text size limitations, we cannot elaborate much more. We agree that the question open new hypotheses to be further tested, e.g. using this type of modelling approach. However, a stochastic model would then be more convenient to question the impact of refuge / buffer zone and spillover, as population size becomes locally very small and thus random processes could have a huge impact on population dynamics.

a few minor points follow:

line 10 should be 'and thus'

Done

line 27 should be 'and helps us to'

Done

line 55 is it the decrease of the temperature over the period between larvipositions that matters, or is it mean here that the period is lower at lower temperatures?

This has been rephrased to: “Between 20-30°C, the lower the temperature is, the longer the period between larvipositions” (L60).

line 58 should be capital 'C' for temperatures (2x)

Done

line 61 'to their first host'

This sentence has been modified (L66-8): “Acquired feeding preferences have been demonstrated in this species, i.e., the host selected for the first blood meal can influence the one selected for the second meal.”

line 101 how was the grid cell size chosen? It looks like the cell size is well below the scale at which fly population effects typically vary (at least in the figure) but it would be helpful if this relationship were quantified, e.g by something like a measure of the variation of key quantities within a cell, compared to the variation between cells. if the former is high compared to the latter, it would suggest you might be missing some important effects

The cell size was chosen according to the radius of fly dispersion over a day (250m). The within-cell variability of population size is much smaller for a vast majority of cells than the between-cell variability over time, which is now mentioned in the text (L182; see also Fig. S7 in SI).

line 196 the lack of importance of temperature doesn't seem intuitive to me and is therefore interesting - it would be helpful to have a bit more discussion this.

After a homogeneous control, two criteria are used to assess the control: (1) the contribution of cells to the total population size, which typically varies with the carrying capacity (Fig. 2B); (2) the local impact of control in cells, which typically varies with temperature (Fig. S12). During the heterogeneous control, cells to be controlled are chosen using score P (SI6), which is a synthesis between cell contribution and control impact: cells where the population decreases the most during control are targeted first all the more if they highly contribute to the global population. As a result, cell selection is related to carrying capacity (Fig. S13) as cells of very low carrying capacity do not highly contribute to the global population, but cell selection is not related to temperature (blue and orange dots cover the whole range of temperature). This could explain why the criteria (2) is not correlated anymore to temperature. A sentence has been added in the discussion about that explanation (L252-5).

Appendix B

Manuscript resubmission - RSPB-2020-0749

Dispersal in heterogeneous environments drives population dynamics and control of tsetse flies

H. Cecilia, S. Arnoux, S. Picault, A. Dicko, M.T. Seck, B. Sall, M. Bassène, M. Vreysen, S. Pagabeleguem, A. Bancé, J. Bouyer, P. Ezanno

First, we would like to thank the associate editor and the three reviewers for their helpful and constructive comments, which substantially helped us improving further our revised manuscript. We have addressed (in blue) all their comments (in black) thereafter and have modified the manuscript accordingly. Modifications are highlighted as tracked changes in the manuscript and the SI.

Second, as advised by the editor Pr. Loeske Kruuk, we would like to take the opportunity of this response letter to inform the associate editor that we have discussed the review with the editor and how to respond to it. See a copy of the editor's email in grey below (date: 05th August 2020). He agreed not to send back the manuscript to Pr. Hargrove due to a conflict of interest. You will see that we addressed Pr. Hargrove's comments in details thereafter. When relevant, we modified the manuscript accordingly.

Dear Professor Ezanno,

Thanks for your email. I appreciate that this was a fairly unusual review (and that it only came in on the second round of reviewing for your paper).

The paper will not go back to Prof. Hargrove for another review; it is clear that there is a long-standing difference of opinion there which would be unlikely to be addressed in a revision.

However, please address as much of his comments, or acknowledge the alternative view points, as possible. I appreciate it is a long review, and you could end up writing an excessively long response if you replied to each point in detail. So if it's more efficient to combine responses to some points into a single response, then please do so.

The Associate Editor will see the revision before I do, so it would be useful if you mention in your response letter that you have discussed the review with me and how to respond to it.

Good luck with the revision, and best wishes,

Loeske Kruuk

Associate Editor - Comments to Author

I have now received three reviews including a new one that raises substantial issues concerning the parameterization, assumptions and interpretation of your results. Everyone agrees that the model is very interesting and is a good contribution to the field. You have also done a good job clearing up issues raised by previous reviewers. However, my sense is that the detailed critical review has clearly raised a number of important points that need to be addressed. In my reading of the paper I also found the scattered statements on the effectiveness of SIT somewhat out of place in the manuscript. You are not really addressing

the relative effectiveness of control strategies in any direct way and it was unclear to me why these assertions were in the paper. It may be that you need to directly address this with specific modeling.

We agree and indeed we almost did not mention SIT in the manuscript. We addressed this aspect only in the discussion at three locations, to explain the next steps after a low population density has been reached. To decrease the feeling that effectiveness of SIT is discussed (which is not the case), we modified the discussion.

First paragraph of the discussion, previously we wrote: “Such a heterogeneous impact can be partially compensated during eradication campaigns by releasing sterile males by air that will aggregate in the same sites as wild males, as observed in the eradication campaign against *Glossina austeni* on Unguja Island of Zanzibar.” → as it was the first paragraph of the discussion which intends to give our main findings, we have removed this sentence (previously written to provide practical perspectives).

L312-320: previously we wrote “Tsetse control methods can only be efficient when the ecological strategy of the targeted species is taken into account. Fast action methods (e.g. insecticides) are better suited for species showing high reproductive rates, short generation times, broad food preferences, and good dispersal abilities. In contrast, pests reproducing at low rates with long generation time, but good competitive abilities would be more efficiently managed using host resistance, cultural and sterilization methods. Nonetheless, such characteristics should be considered in conjunction with species relationships in communities.” → this is a general ecological comment, which does not focus on SIT. It intended to broaden the perspective beyond tsetse control. It has been rephrased to: “The field of ecology has theorized in the past that fast action methods (e.g. insecticides) are better suited for species showing high reproductive rates, short generation times, broad food preferences, and good dispersal abilities [72]. And that, in contrast, pests reproducing at low rates with long generation time, but good competitive abilities would be more efficiently managed using host resistance, cultural and sterilization methods. This has already been nuanced, as such characteristics should be considered in conjunction with species relationships in communities [72]. Our study shows that it is time to move past this dichotomy and consider the effect of combined and spatially targeted control measures to achieve eradication.”

Previously we wrote (L330-334 previous version) “However, generation time was also a contributing factor to variations in tsetse population size in our analyses. This is not an easy parameter to modify, but it is the aim of the sterile insect technique (SIT). The SIT impairs reproduction but to be effective, it needs to be applied when population densities have been reduced to a low level.” → when parameters vary over one year (current sensitivity analysis) and not three (former one), mortality becomes the most contributing parameter (over three years, the generation time was also important). Thus, we have removed this sentence and restructured the paragraph to make it clearer.

Many of the other points by the critical reviewer may well be addressed with either rewrites and some modeling. One of the previous reviewers also makes an important point about the need for the spatial model and therefore whether spatial structure is playing a role here. I was unsure what the relative role of space is in the model – clearly there is a big effect of environment and it is realistic to have this varying in space, but is it the case that it is environmental heterogeneity rather than space that matters? How many of your results could be obtained with just the environmental heterogeneity that you show? I think you could give

the readers a good sense of that. Both of these referees make some minor comments that would improve the manuscript.

This is an important question, which deserved additional analyses to be addressed. See our detailed answer to Referee #2 on that question. Based on our new results, we can now conclude that - even for moderate dispersing species such as tsetse flies in fragmented habitat - there is a clear need for spatial models to assess targeted control strategies and their consequences on the population dynamics after control has stopped.

In terms of the papers' suitability for PRSB, you need to address the issues of the critical reviewer, and also be clear why the paper is more than a specific model showing how environmental factors may impact the control of a particular pest. The paper may make a general impact in pushing the approach of using data driven environmental parameterization of vector control models, but it may have a bigger impact in a more focused journal. You need to be clear about the general insights from the model beyond the system.

Our findings are of broad interest, i.e. outside the "tsetse world", as we clearly highlight the importance of the spatial component in population dynamics even when individuals disperse over short distance (here one cell). In addition, we illustrate that using a model applied to a real pest management issue. To our opinion, this is an important and not straightforward take-home message. We have updated the paper title and main text to better reflect this finding.

Reviewer(s)' Comments to Author

Referee: 2 - Comments to the Author(s)

The paper is clearer and the previous comments addressed. I do have one additional concern that has arisen from the revised manuscript. As it stands, it is not clear as to the extent to which the spatial element of the model is important. The grid squares that the authors have used seem to simply divide the population into two areas, some viable, some not, with a boundary layer in between. This is not an issue in itself, however what it does not address is whether or not the persistence of vector populations is locally driven (that is, by local parameters) or whether the interaction between the cells is driving persistence. There are hints in the paper but it is not discussed explicitly and it would be useful to have this discussion. In particular if it is the local within-cell scale that is most important, then a spatial model is arguably not necessary at all (just a map of local conditions). However if between cell persistence is maintaining the population, then the need for the spatial model is clear.

This is an important question, which deserved additional analyses to be addressed. To highlight whether or not the persistence of vector populations is driven by interactions among cells, we compared model predictions assuming spatial tsetse fly dispersal vs. assuming no dispersal. We obtained very interesting results.

With regards to control efficacy, when applying a homogeneous control, efficacy was similar if neglecting or not fly dispersal, but with a lower effect of temperature when neglecting dispersal and less variations among cells (L215-216, Supporting Information 7, Fig. S14-S15). In such a case, the spatial component is thus not that crucial. In contrast, when applying a heterogeneous control, efficacy was much less variable when neglecting dispersal.

Temperature and its variations impacted control efficacy as in the homogeneous case (L231-234, Fig. S16). This indicated that a non-spatial model largely misestimates the efficacy of targeted control and may lead to different conclusions.

With regards to resurgence, the difference was even more striking. While there was no difference in resurgence if considering or not dispersal after a homogeneous control (L244, resurgence is very low in both cases), results largely differed after a heterogeneous control. Neglecting dispersal (i.e. no recolonization is possible) lead to similar results to the homogeneous case (L251-254), while resurgence was very high with dispersal, especially at the frontier between the controlled and the uncontrolled areas where the local population could be multiplied by 6 to 44, with no effect of temperature.

Based on these new results, we can conclude that - even for moderately dispersing species such as tsetse flies in fragmented habitat - there is a clear need for spatial models to assess targeted control strategies and their consequences on the population dynamics after control has stopped. To our opinion, this is an important and not straightforward take-home message, thus we updated the paper title to better reflect this finding. Abstract and main text have been updated accordingly at several locations. All the figures concerning control and resurgence and assuming no dispersal are provided in the supplementary material (especially in SI §7).

A few minor points below:

In the abstract should be 'variations' Done

line 24 should be US\$ Done

line 47 should be 'increases' Done

line 71 - why concentrate on gambiensis.

This subspecies is currently targeted by several control programs in West-Africa and is an important vector of human and animal trypanosomoses (see Bouyer 2020 for details).

line 285 "variation" Done

line 88 onwards - some references supporting these statements would help especially for the broader audience of the journal.

This now writes (L93-96): “In tsetse, the physiological age structure is an important feature which depends mainly on the survival rate and the length of the pupation period, both conditioned by temperature [16]. When the temperature drops (cold dry season), pupation lengthens, adult flies live longer, and the population becomes older [47]”.

line 99 - where are the trap catch data described? and would it also have seasonal dependence?

Trap catch data are described in SI section 2.2. We added more details to make it clearer.

Also, capture-mark-release (CMR) data used for calibrating dispersion, trap catch data used for assessing apparent density (as well as data on physiological age) are now provided as CSV files in the Sourcesup folder (and the ReadMe has been updated). SI section 8 lists data files provided.

We are not fully sure to understand the point on seasonal dependence. Trap catches were conducted homogeneously through seasons. It is known that dispersal of tsetse increases in the rainy season thanks to a higher relative hygrometry, thus reducing trap catches in their preferred habitats, but this is not related to a variation of trap efficiency.

Referee: 3 - Comments to the Author(s)

Overview: This paper has some very good elements and should definitely be published in some form.

Model: The model itself looks good and is a useful addition to the literature.

Input parameters: The underpinning of the model, however, has not been given sufficient attention and it is not clear to me that the authors have actually got correct values for some of the input parameters. Whether or not I am right in this regard, the reporting of the estimation of those parameters – in the Supporting Material – leaves a lot to be desired. I would say that 99% of readers would be completely unable to fathom how the authors have arrived at their estimates for the input parameters. In an attempt to discover what the authors have done I have, therefore, spent a lot of time and effort in unpicking all of the details.

We agree that in some places, explanations were lacking on the calculation of input parameters. We extended the SI to provide more explanations as required, except when input parameters came from published data (which were cited). Also, we now provide access to capture-mark-release (CMR) data used for calibrating dispersion, to trap catch data used for assessing apparent density, as well as to data on physiological age, as CSV files in the Sourcesup folder (the ReadMe has been updated). SI section 8 lists data files provided. We realized during the revision process that the equation for pupal duration was slightly incorrect, because of an issue in the allocation of control groups. We provide thereafter more information on specific calculations. We re-run the model and to reproduce all of the results with corrected values for impacted parameters (Table S2). It did not change at all our results and conclusions. The figures have all been updated in main text and SI, including those related to the sensitivity analysis. To save time, the sensitivity analysis has been done over one year of simulation instead of three, which was more in line with the rest of the paper. Interestingly, it puts more emphasis on the model sensitivity to variations in adult mortality, which explains over a year more than 50% of the output variations. It reinforces the focus on that issue and is in line with our other results. It made the flow of the paper clearer. As a result, we remove Table S4 (see the updated version thereafter) from SI as it was useless. We updated the figures to be shown focusing only on varying outputs.

Table S4 (not in the paper anymore). First order sensitivity indices by input parameters, for varying outputs, by decreasing order of percentage of variance explained. Only the first four input parameters are shown. See definition of outputs in Table S3.

Varying outputs	$\mu_{\{N,F,M\}}$	% of variance explained		
		2 nd	3 rd	4 th
popMean	52.9	12.2 (θ)	11.4 (δ_P)	7.6 (δ_N)
popStd	51.3	13.9 (θ)	10.5 (δ_P)	8.2 (k)
surf50	52.1	12.6 (δ_P)	8.2 (k)	7.8 (δ_N)
mean_distrib	52.2	12.6 (δ_P)	9.5 (k)	7.3 (θ)
q5_distrib	50.8	11.4 (δ_P)	8.5 (δ_F)	8.5 (θ)
q25_distrib	52.7	12.4 (δ_P)	7.9 (δ_F)	7.8 (δ_N)
q75_distrib	50.3	12.7 (δ_P)	11.9 (k)	6.7 (δ_N)
q95_distrib	51.0	13.0 (δ_P)	12.1 (k)	8.5 (θ)
median_distrib	51.3	12.4 (δ_P)	8.9 (k)	7.5 (δ_F)

We consider that the full paragraph below highlighted in yellow is completely wrong since we provided strong evidence supporting the claim that environmental heterogeneity drives tsetse populations dynamics. We cannot answer point by point to this paragraph since it is more related to a political position (for chemical control and against SIT) rather than on scientific facts. We would like to point out that currently, tsetse flies have been eradicated in less than

2% of their originally infested area (Bouyer et al. 2013). This is mainly because control programs have been based on chemical control with insufficient knowledge of the ecology and dynamics of the target populations. Efficient integrated pest management strategies rely on a good knowledge of the target populations [14]. This is true for most of insect pests and not to be debated anymore. In the particular case of tsetse, the reviewer cites examples that are mostly not relevant:

- campaign (i) relied on aerial spraying of persistent insecticide which is not permitted anymore because of its environmental impact (see: Ciss et al. 2019, Bouyer et al. 2018): of course, this kind of campaigns were successful in the past but belong to the past;
- campaign (ii) was also conducted with ground spraying of residual insecticides; it must be noted that all the cleared areas were lost thereafter thanks to resurgence/re-invasion, see <https://www.sciencedirect.com/science/article/pii/S0001706X19313671> ;
- campaign (iii) used ground spraying of residual insecticides. It was successful and the cleared areas are tsetse-free until now thanks to an extension of agriculture that destroyed the suitable vegetation but this method is not acceptable anymore;
- campaign (iv) was successful but most of the cleared areas have been re-invaded ;
- campaign (v) was successful because it addressed the Okavango delta, a very homogeneous area with open savannah, with a very homogeneous control method, e.g. sequential aerosol spraying of insecticides applied by plane: this method failed against *Glossina palpalis gambiensis* in Ghana [75] and we recently explained why (De Deken & Bouyer 2018).

It is clear that environmental heterogeneity played a huge role and this example provided by the reviewer fits very well to our conclusions. It must be specified that Dr. Hargrove worked mostly on savannah tsetse species in open environments, which explains largely his misunderstanding of our conclusions on riverine tsetse flies living in fragmented environments. However, his detailed comments below were very useful to improve the paper and were all considered in the new version, see our replies below.

Claims: Finally, it has to be said, that the evidence presented does not actually support the claim of the title that “Environmental heterogeneity drives tsetse fly population dynamics and control”. This idea leads on to further indefensible statements such as:

Lines 295/296: “Tsetse control methods can only be efficient when the ecological strategy of the targeted species is taken into account [73].”

Lines 296/298: “Fast action methods (e.g. insecticides) are better suited for species showing high reproductive rates, short generation times, broad food preferences, and good dispersal abilities [74].”

If the first claim were true, it would suggest that control operations against tsetse where ecological strategies have not been taken into account would likely be unsuccessful.

Moreover, since tsetse have relatively low reproductive rates, long generation times, narrow food preferences, and somewhat limited dispersal abilities, the second claim is clearly suggesting that insecticidal control measures are likely to be ill-suited for tsetse control and likely to be inefficient and unsuccessful.

Here are some counter-examples to those claims. To the best of my knowledge, in none of the examples below was the “ecological strategy of the targeted species ... taken into account”. Nonetheless, the operations were extremely efficient and, more to the point, were successful in achieving eradication:

- (i) The eradication, in the early 1950s, of *G. pallidipes* from South Africa using aerial spraying.

du Toit, R. (1954) Trypanosomiasis in Zululand and the control of tsetse flies by chemical means. Onderstepoort Journal of Veterinary Research 26, 318-385.

(ii) The eradication, in the early 1960s, of *Glossina tachinoides* and *G. morsitans* submorsitans from parts of Nigeria.

Davies, H. (1964). The eradication of tsetse in the Chad river system of Northern Nigeria. Journal of Applied Ecology 1, 387-403.

(iii) The eradication, in the early 1980s, of *G. m. morsitans* from the Umfurudzi Wildlife Area in Zimbabwe.

Reported in Hargrove, J. W. (2003) Tsetse eradication; sufficiency, necessity and desirability. DFID Animal Health Programme, Edinburgh, UK. 133 + ix pp.

(iv) The eradication, in the 1990s, of *G. m. centralis* and *G. pallidipes* from large proportions of Bukoba and Karagwe Districts in Tanzania.

Hargrove, J. W., Omolo Silas, Msalilwa, J.S.I. & Fox, B. (2000) Insecticide-treated cattle for tsetse control: the power and the problems. Medical and Veterinary Entomology, 14, 123-130.

(v) The eradication, in the 2000s, of *G. m. centralis* from the whole of Botswana, specifically the Okavango Delta and adjoining areas.

Kgori, P. M. Modo, S. & Torr, S. J. (2006) The use of aerial spraying to eliminate tsetse from the Okavango Delta of Botswana. Acta Tropica, 99:184–199.

None of these eradication campaigns is referenced in the paper under review. Instead we are told Lines 298/299: “In contrast, pests reproducing at low rates with long generation time, but good competitive abilities would be more efficiently managed using host resistance, cultural and sterilization methods”.

At that point, and given the counter-examples cited above, the paper under review is unmasked as (yet another) vehicle for selling the virtues of SIT, and to diminish – and indeed ignore completely and deliberately – all of the evidence indicating that SIT, as applied to tsetse, is an unnecessary, inefficient and hugely costly approach to eradication. Most amazingly, the MS denigrates insecticidal methods as being unsuitable for the slow breeding and long lived tsetse, while claiming that SIT is best suited to such creatures. The authors have got their ideas entirely wrong. In fact, it is common sense that SIT is best for insects that reproduce fast and live briefly, and SIT is seriously inappropriate for the slow breeding and long lived tsetse. This is especially so since it is the longlived adults of both sexes that transmit disease directly, rather than via their eggs and larvae.

Let me close by addressing a comment to the first author – who is clearly a young, up-and-coming, talented modeller who should not be discouraged by the criticisms of an elderly curmudgeon. Dr Cecilia, I mean it when I say that I like the modelling part of this paper. And I hope you will not be discouraged by the disparaging things I have had to say about other parts of the paper. In over 50 years of work on tsetse I have, myself, had more rejections of papers than I care to remember. And I know that these rejections, or even criticisms, can be very dispiriting. However, take heart. There is much hope for this paper and I wish you well – my comments would not be as detailed if I thought otherwise. My advice is simply this: keep to the science. Get the science right and the rest will follow. Leave the nonsensical geopolitical grandstanding to the denizens of Vienna.

No comment.

John Hargrove, Stellenbosch, 3 June 2020

Detailed comments

Model

Lines 63/64. The authors quote Randolph & Rogers (1991) in claiming that: “Temperatures below 20°C or above 30°C increase adult mortality [44].” That is not what Rogers & Randolph say. Of course, if one keeps flies in a deep freeze they will die. But I do not know of evidence showing that adult mortality of tsetse – in the field – increases at temperatures below 20°C. For what it is worth, Hargrove (2001) used mark recapture data for island populations of tsetse to estimate that survival probability increased with decreasing mean temperatures down as far as 16°C.

Hargrove, J. W. (2001) Factors affecting density-independent survival of an island population of tsetse flies in Zimbabwe. *Entomologia Experimentalis et Applicata* 100, 151-164.

We agree with this and the statement has been corrected. This was actually a mistake since our model assumes a constant mortality below 20°C. Thanks for pointing this out.

Lines 86/88. It is claimed that: “... the population age structure was calculated using natural abortion rates (Supporting Information 2.3) that were compared with simulation results for qualitative evaluation of the model.” Scrutiny of Supporting Information 2.3 reveals, however, that – as with the estimation of pupal duration – we are given scant detail about how the age structure was calculated. The authors should show, in detail, how they estimated the abortion rate – giving details of the age-structured sample sizes used for this exercise. Then they should tell us how they used this abortion rate to generate the population age structure. I do not know of any example where this method has been used in the past. Finally, the authors need to say how they handled the fact that traps provide samples of tsetse that are biased with respect to age. How did they estimate the size of this effect, and allow for it in making their calculation of the true age structure? These matters should be addressed before the paper can be published.

We agree with this comment. The sentence was misleading, and probably the result of a rewriting error during the revision process. We rephrased, lines 90-91: “Field data on population age structure was available thanks to a study focusing on natural abortion rates (Supporting Information 2.3).” We provide details on the protocol for estimating the population age structure and abortion rates in SI (section 2.3). We used the protocols described by Challier (1965).

We are aware of the sampling bias of traps towards non teneral flies and fly > 7 days old, and that is why we used F1 to F3, but not nulliparous females, to check model predictions (Fig S6). This is written L153-155: “Predicted age structure was compared with field data for females of ovarian age 1, 2, and 3, because traps do not catch nulliparous and females of ovarian age 4 and 4+ as efficiently as females of intermediate ovarian ages [65].”

Lines 176/177. It is stated that: “..... carrying capacities were highly heterogeneous and local densities ranged between 112 and 104,768 flies per km² (median: 2,320).” I find entirely ludicrous the claim that any 1 sq km cell in the study area concerned could have a carrying capacity >100,000 tsetse. How did the authors arrive at this number? In lines 94/99 we are told: “... local carrying capacities (Fig. S1C; Supporting Information 3.1) were defined as the maximum sustainable number of tsetse fly individuals per cell (the model’s spatial unit), estimated as $SI \cdot ADT / \sigma$, where SI is the suitability index (estimated with a species distribution model [40] based on maximum entropy), σ is the trap efficiency (probability that a trap catches a fly within 1 km² within a day [59]), and ADT is the apparent density of the fly population (flies sampled per trap per day) [60].”

While we are referred to Fig. S1C and Supporting Information 3.1, this leaves us none the wiser – because we are not given any data on trap catches, nor are we told about the assumed value of the trap efficiency, nor how, or whether SI, and ADT are related. So, once again, we have to work out for ourselves how it will be possible to get carrying capacities of >100,000. We can presume that SI takes values between 0 and 1, as defined by Dicko et al (2014) and, presumably, for the highest carrying capacities $SI = 1$. From Dicko et al (2014) we are also told that: “In the Niayes, observed trapping efficiencies were as low as 0.3% per km² per day ...”. Putting these numbers into $SI \cdot ADT / \sigma$ we see that, to generate a carrying capacity of 104,768 flies per sq km, traps would have to catch $(104768 \times 0.003) / 1 = 314$ flies per trap per day. This is about 150 - 300 times as high as the ADT values prior to control (Dicko et al (2014, Figure 5). How do the authors reconcile these numbers?

First, we would like to point out that, while our carrying capacities were expressed in number of flies per km² to use a common unit, the cells in our grid are 250m x 250m (1/16 of a km²). The highest carrying capacity obtained in one cell is 6,548. However, at no point do we reach >100,000 flies per km² in our simulations. Indeed, as can be seen on top of Figure 2B1, the total number of female flies in the whole grid (~56km²), at the end of the reference scenario, is 181,306.

The suitability index was computed with a complex geostatistical model, and we refer the reader to the associated publication (Dicko et al. 2014) for more details. The trap efficiency was calculated from extensive Mark-Release-Recapture (MRR) conducted during more than a year in four sites and are very robust. Its value is now stated L102 “probability that a trap catches a fly within 1 km² within a day [58], set to 0.003 [39]”.

Concerning capture data for apparent density, the maximum number of flies trapped in a day within 1 km² is 223 in the Niayes, which is consistent with what the reviewer calculated. The slight difference can be explained by the extrapolation performed to fully cover the grid, as we cannot have field data for all the pixels. It has to be noticed that this range of values is reached in a very few pixels, the interquartile range of carrying capacities being 29.4 - 514.9 (pixel scale), which corresponds to 470.4 - 8,238.4 per km².

Capture-mark-release (CMR) data used for calibrating dispersion and trap catch data used for assessing apparent density are now provided as CSV files in the Sourcesup folder (the ReadMe has been updated). SI section 8 lists data files provided.

The paper referred to as support for the Maxent method is, again, Dicko et al. (2014). But that paper appears only to predict the suitability of a piece of habitat for tsetse. It does not pretend, as far as I can see, to be able to come up with estimates of the carrying capacity. If I am wrong in this regard, I will be much interested to see how this is done – and the authors certainly need to provide readers with this information.

To see how we used habitat suitability to predict apparent densities, we referred to Dicko et al. 2015 [59]. It is another work than the one cited above.

I wonder, by the way, whether the authors have reflected that 100,000 flies, each taking order 20mg of blood a day will require about 60 kg of blood per month to survive. What was the host density in the area? In this regard, what concerns me about the Maxent method of estimating carrying capacity is that, as I understand it, the method is basically looking at suitability from physical, vegetational and climatological standpoints. But an area could fulfil these criteria brilliantly and still have a carrying capacity of zero for tsetse if there were no hosts present. This issue needs to be addressed and discussed.

First, at no point do we reach 100,000 flies per km² in our simulations. Indeed, as can be seen on top of Figure 2B1, the total number of female flies in the whole grid (~56km²), at the end of the reference scenario, is 181,306... Also, beware that favourable patches are far smaller than 1 km², thus there is nowhere this absolute amount of flies locally.

By the way, hosts are not limiting for tsetse flies in this area where human and cattle density are very high (Bouyer et al. 2014). This is now clearly stated L120: “We consider the flies not limited in their access to hosts”. Riverine species can feed on a large variety of hosts from lizards to human and domestic cattle. Their densities can thus be predicted based on physical, vegetation and climatological data derived from satellite images, as originally demonstrated by Rogers & Randolph [43], thereafter refined based on phyto-sociological analyses ([52, 54], Bouyer et al. 2006), and culminated in [59].

Lines 189/190. It is stated that: “A 5% variation in temperature resulted in demographic explosion or extinction, ...”. The authors need to provide more detail here. On the face of it, this seems like an extravagant claim. Thus, imagine that a population “explodes” at 25°C. A 5% (or 1/20) variation would change this temperature by about 10C. Are the authors saying, then, the population would go from “explosion” at 25°C to “extinction” at either 24°C or 26°C? The authors should clarify this matter.

We have to remind here that temperature is varying over time and in space, ranging in our study area from 18.7°C to 29.9°C. The tested variation of 5% impacts all temperatures simultaneously (in space and time), keeping similar the pattern over time and the relative variations in space. When decreased by 5%, low temperatures become even lower and medium temperatures become low. Thus, there are fewer areas favourable for tsetse development leading to a negative population growth rate and - after a while - to population extinction. This is why there is no endemic tsetse population in areas where the temperature is too low to sustain populations. With 5% higher temperatures, we stay in a range compatible with population maintenance but at a higher population density. The word “explosion” was not appropriate and we reformulated the sentence L193-195: “A 5% variation in temperature resulted in significant population increase or extinction, largely outweighing the effect of a similar variation in carrying capacity (Fig. S9), which emphasised the need for considering reasonable temperature variations.”. A similar observation has been made in southern Africa by Moore et al. [69]. It is also important to keep in mind that the temperature experienced by flies is “are normally 2-6°C cooler than the ambient temperature [60]” (line 106). Our results therefore imply that the ability for flies to find cooler resting places is crucial for their survival.

Input parameters

I am concerned about the evidence regarding various rates of development, as presented in the Supporting Material 2.1 Mortality, fecundity, and length of the pupal period. In particular, we need to be given much more detail about the estimation of the pupal duration. Above all, we need to be told explicitly how the experiment was done. Is it correct that all pupae were held at 25°C for 20 days – and then held, during phase 2, at the test temperature until they emerged? The reader can perhaps infer that this was the case – but should be told, rather than having to guess. The authors should also make clear that the group of 120 initial pupae was an undifferentiated collection of males and females. Thus the total number of flies emerging, out of the 120 initial pupae is the sum of the males and females for that experiment. Assuming that the above interpretations are correct a number of questions still arise about the presentation and methodology.

These interpretations are correct. We tried to make it clearer in the revised version.

- (i) Where do the “development rates” come from in Phase 1? Since the pupae are all 20 days old, and have not yet hatched, how do the authors know the rate of development? My guess is that they don’t know at all. They have assumed that the rates at 25°C are the same as those estimated by Phelps & Burrows (1969) [henceforth P&B] – see P&B, page 35, Table II. But we are not told this: we have to guess, or work it out for ourselves. [Aside: I note that a careful check shows that the authors have actually used the wrong numbers from P&B – though, serendipitously, the resulting error is very small. Thus, their assumed rates for males and females are 0.0345 and 0.0369 for male and female pupae, respectively. These are not the rates quoted by P&B – they are actually 1/100 of the quoted pupal durations for females and males, respectively. That is to say, the authors have: (a) got the sexes the wrong way around and (b) used pupal durations/100 instead of rates. Happily for them, the correct rates (given below Table II in P&B) are actually, and as I say serendipitously, very close to those assumed by the authors – 0.0344 for males and 0.03683 for females. End of aside].

We had a control group that stayed at 25°C for all the duration of the pupal development which allowed us to calculate precisely the development rate at 20 days, and thus adjust in the other groups. It has been specified in the new version (in Supplementary Information 2.1 and Table S1).

The difference between male and female pupae was very small, and we considered in our study similar pupal duration for male and female pupae.

- (ii) Where do the rates come from in the last column of Table S1, Phase 2? Again, I THINK I can work it out. But the readers should not have to do this: the authors should spell it out explicitly. Here is what I think happens – using females as an example. Using the assumed development rate of 0.0369 per day in Phase 1, the authors estimate that a 20 day-old female will have completed $20 \times 0.0369 = 0.738$ of development, meaning that there is $1 - 0.738 = 0.262$ remaining. They then divide this number by the mean number of days spent at the test temperature before the adult emerges, to get the development rate at the test temperature. For example, in the first of the three experiments at 20°C the mean time to hatching was 19.19 days and we have a rate of $0.262/19.19 = 1.37 \times 10^{-2}$, the value quoted in Table S1.

We did not assume the development rates at 25°C to be similar to the ones found by Phelps and Burrows. We calculated them using the control groups of our experiments, which we clarified in Supplementary Information 2.1 and Table S1. In Fig. S2B for pupa development, only the orange line, which corresponds to Hargrove’s equation (2004), is based on data from Phelps and Burrows (1969).

- (iii) Where do the pupal durations come from in Figure S2B? Are the readers meant to infer that they get this from the inverse of the daily development rate? If so, why not tell them? And, if not, the authors need to say exactly how they do arrive at the numbers in question. I presume the duration is estimated by the inverse of the rate, so that - for the above example – the duration would be $1/0.0137 = 73.0$ days

Yes, duration = 1/rate, as very classically done. It is now mentioned in the SI.

- (iv) I note that, while the estimated pupal duration at 25°C are closely similar to those measured directly by P&B, the above-cited pupal duration of 73 days at 20°C is >50% higher than the 47 days observed by P&B.

Indeed, our equation is close to the published one for *G. morsitans* for temperatures above 25°C, but the sigmoidal function adapted to the data we have for our species implies a longer duration at a temperature around 20°C and shorter durations for colder temperatures.

- (v) By contrast, the estimated duration at 15°C is 16% lower than the figure estimated by P&B. The authors make no comment about these differences.

There is nothing to comment, this is not under the scope of the present study. The idea was to use values adapted to our species, when enough data were available to derive new estimates.

- (vi) But these differences really must be resolved. On one hand, if the results for *Glossina palpalis gambiensis* are correct, then they suggest that there are marked differences between this species and *G. m. morsitans* – not only quantitatively, but also qualitatively, since the shape of the functions linking pupal duration and temperature appear to take an entirely different form in the two species. If this were true, it would be a very interesting result – and worthy of a paper in its own right. Since the co-authors have access between them to: (i) pupae and (ii) incubators, there is every opportunity for some keen PhD student to make a proper job of estimating pupal durations for several riverine species – taking the sort of care that R.J. Phelps took with *G. m. morsitans*.

Indeed, it was for a different species, and it was not the objective of the present publication to go further in this analysis. However, we thank the reviewer for his suggestion and will take it into consideration to plan future studies.

- (vii) As indicated above, I have to say that I have serious questions about the reporting and methodology in the current study, as I shall now detail.
- (viii) To estimate the pupal duration at 15°C it is necessary to alternate the incubation conditions between 15°C and 25°C – otherwise all of the flies die (P&B). However, this is not the case for temperatures >20°C and ≤32°C. At these higher temperatures, it is unnecessary to keep the flies at 25°C for 20 days, prior to keeping them for the remainder of the time at the test temperature. It is also undesirable, because in so doing one is forced to make approximations regarding the assumed rates of development at 25°C and at the test temperature. Instead, the flies should be kept at the test temperature for the whole pupal period. That way one measures the period directly.

We agree with you and we had a control group at 25°C for the full pupation duration, as specified above.

- (ix) Given the very large differences between the results under review and those of P&B, I think it is imperative – before this paper is published – that the authors take 120 more pupae and incubate them at a constant temperature of 20°C, throughout their pupal period. This will immediately determine whether the difference between the result under review and the P&B results are real – or whether they are an artefact of the slightly different methodologies used in the two studies.

This was already done, but at 25°C.

- (x) The major methodological difference was that when P&B wanted to estimate rates of pupal development at low temperatures ($T < 16^{\circ}\text{C}$) they alternated the incubation temperature every 24 hours, between 25°C and the test temperature. This is NOT what the authors have done here: they have instead kept the pupae at 25°C for the first 20 days of adult life.

See our replies above.

- (xi) Phelps was very careful to check that his approach gave reasonable results by first doing a test where he alternated pupae at temperatures of 20 and 25°C, respectively. He was, thereby, able to check that he got the same estimated pupal duration using this indirect method as he got when he simply kept the pupae at 20°C throughout their pupal period.

Yes, we did exactly the same, but at 25°C...

- (xii) The present authors have failed to carry out this due diligence – and should definitely do so before they publish this paper.

No, but we agree that it was not well explained. It is done now.

There are four reasons that I have taken so much time and trouble to tease out the problems with the estimation of pupal duration presented in this paper. (i) I object to sloppy work. (ii) General interest: I have emphasised in the past that we do need to get accurate estimates for development rates in tsetse species other than *G. m. morsitans*. (iii) P&B noted that at low incubation temperatures, pupal mortalities rose rapidly. This matter is not discussed in the present paper – and does not seem to be reflected in the modelling. (iv) Finally, it is particularly important for this paper that all developmental rates are accurately measured at low temperatures, because the authors attach importance to the claim that “the coldest cells with the smallest temperature variations acted as refuges when adult mortality was homogeneously increased, control being less effective in such refuges”. This claim will be thrown into deep doubt if it transpires that, for example, there are large errors in the estimates of pupal duration at low temperatures – and that they have not taken into account increased pupal mortality at low temperatures.

First, there are no large errors in our estimates of pupal duration. The work has been done properly. Furthermore, accounting for an increase in pupal mortality at low temperature is important if you encounter low temperature and thus fully depends on what you call low temperature! In our study area, the minimum temperature locally is 18.7°C, which is not that low. In addition, 75% of the local temperature values are higher than 22°C. There was thus no need to study pupation at a lower range than what we did. We did not reach the temperatures leading to increased mortality in P&B studies.

Other issues with Supporting Material 2.1: Equation S1 is provided with no explanation about what each of the variables refers to, no justification, nothing. What is the equation all about? What is it used for? Mortality, fecundity, pupal period, or something else entirely?

This equation is now provided together with the other equations to facilitate the reading. Reference is made to the associated figure and to the table showing the parameter values.

Referee: 4 - Comments to the Author(s).

This manuscript presents thoroughly a deterministic compartmental model of tsetse fly population dynamic that accounts for a spatially heterogeneous environment (through a regular grid of 900 cells), movement of flies between cells and a density dependence of the fly dynamic. It includes a sensitivity analysis of the model input parameters on population size that demonstrates that the population dynamic is mostly driven by temperature and adult mortality. Simulations were subsequently conducted to evaluate the impact of an increase of the adult mortality on the reduction of the population, assuming either a spatially homogeneous control strategy or a heterogeneous control strategy targeting the cells with the highest carrying capacity.

I was one of the reviewers of a previous version of this manuscript when it was submitted to Peers Community in Ecology, and have already made my major comments at that time. I found the current version very strong even if it can be sometimes difficult to follow between the main text and the supplementary materials. But I am aware of the length restrictions, so I think it is fine as it is.

I only have minor comments to be addressed before the manuscript can be accepted for publication:

Abstract: From “The coldest cells” to “related to temperature” is difficult to follow and to understand as the sentences are too long. Consider rephrasing them.

We have reformulated the abstract to make sentences clearer and shorter: “The coldest locations with the smallest temperature variations acted as refuges when adult mortality was homogeneously increased. In these refuges, control was less effective and population recovery after control was faster. In contrast, a heterogeneous control, where the choice of targeted locations is optimized, resulted in a similar population decrease but induced more dispersed individuals. Control efficacy was no longer related to temperature. Population dispersal created refuges on the interface between controlled and uncontrolled zones, where resurgence after control was very high.”

L87: I assume “that” refers to “population age structure” and not to “abortion rates”. Please clarify.

True, we rephrased.

L120-121: Suggestion “We used a constant rate to model pupa mortality, given the lack of data on this parameter [31]”. Otherwise it is not clear what this constant rate is referring to.

Rephrased.

Figure 1: l, nP, nN and nF should be defined in the legend. Also consider including “ $S \in \{P, N, Fx, M\}$ ” when you first mention “for stage S”.

Done

L156: “to 5% and 2%”

Done

L170-178: you present Figures S1A and S2B but not Figure S2C. Talking about S2C, given the amount of observed data that were used to calibrate the model of the time to larviposition and the fact that they all relate to 25°C, the relationship with the temperature seems very far-fetched to me. It seems that there is no evidence to justify this association. You should probably consider the time to larviposition as a constant, equal to 18-19 days for nulliparous and 10 days for parous females. Just like you did for pupa mortality.

Indeed, we used the only published equation (fitted for *G. pallidipes*, in dark orange in Fig. S2C) and just checked that the few points we had for our species lie within the confidence interval of this function. This is stated in the legend of Fig. S2: “The few new data available for the time to larviposition were consistent with Hargrove’s equation, which thus was used.”

L210: “smaller increase in mortality”

This sentence was not clear. We rephrased it (L217-219): “Targeting cells contributing the most to population management (greatest carrying capacity and most impacted by an increased mortality) could achieve a similar decrease in population size as a homogeneous control, but required an additional effort in increasing mortality in targeted cells.”

LL323-327: I suggest you include a discussion on the feasibility of implementing heterogeneous control (as opposed to homogeneous)

Indeed, it almost impossible to do a homogeneous control in a fragmented habitat. The traps are usually spread heterogeneously in the habitat (see Dicko et al. 2014, Fig. 4). Our model is thus very useful to highlight in which zones such a trap-based control should be implemented. An interesting perspective would be to predict how many traps per hectare should be installed in these zones, according to their characteristics and the level of mortality targeted locally.

This is now discussed in the paper (L329-332).

SuppMat 2.3: please include the corresponding reference.

This is not already published. We now provide the data as CSV files in the Sourcesup folder (the ReadMe has been updated). The link to this permanent repository is in SI section 8. Files concern: capture-mark-release (CMR) data used for calibrating dispersion, trap catch data used for assessing apparent density, and data on physiological age. SI section 8 also lists data files provided.

Complementary references (those with numbers are listed in main text)

- Bouyer, F., et al. (2014) Ex-ante cost-benefit analysis of tsetse eradication in the Niayes area of Senegal. *PloS Negl. Trop. Dis.* 8, e3112
- Bouyer, J. (2020) *Glossina palpalis gambiensis* (Tsetse Fly). *Trends Parasitol.* in press
- Bouyer, J., et al. (2006) Mapping African Animal Trypanosomosis risk from the sky. *Vet. Res.* 37, 633–645
- Bouyer, J., et al. (2013) Community- and farmer-based management of animal African trypanosomosis in cattle. *Trends Parasitol.* 29, 519-522
- Bouyer, J., et al. (2018) The ethics of eradicating harmful species – the case of the tsetse fly. *Bioscience* 69, 125–135
- Challier, A. (1965) Amélioration de la méthode de détermination de l'âge physiologique des glossines. *Bulletin de la Société de Pathologie Exotique*, 58, 250-259.
- Ciss, M., et al. (2019) Environmental impact of tsetse eradication in Senegal. *Scientific Reports* 9, 20313
- De Deken, R. and Bouyer, J. (2018) Can sequential aerosol technique be used against riverine tsetse? *PloS Negl. Trop. Dis.* 12, e0006768
- Van der Vloedt, A.M.V. and Barnor, H. (1984) Effects of ionizing radiation on tsetse biology. Their relevance to entomological monitoring during integrated control programmes using the sterile insect technique. *Int J Trop Insect Science* 5, 431-437
- Williams, B.G., et al. (1990) Tsetse fly (Diptera: Glossinidae) population dynamics and the estimation of mortality rates from life-table data. *Bull. Entomol. Res.* 80, 479-485

Appendix C

Manuscript resubmission - RSPB-2020-0749

Dispersal in heterogeneous environments drives population dynamics and control of tsetse flies

H. Cecilia, S. Arnoux, S. Picault, A. Dicko, M.T. Seck, B. Sall, M. Bassène, M. Vreysen, S. Pagabeleguem, A. Bancé, J. Bouyer, P. Ezanno

Associate Editor - Comments to Author

Thank you for your detailed replies to the referees and the excellent revision of the paper. I think it is much improved and you have made positive changes in response to the critical referee where you can. The two referees who have looked at this version are positive. There are a few relatively minor comments from one of the referees for you to consider that will further improve the manuscript. I enjoyed reading this version of the paper.

Thanks for the time spent on our paper!

We added a media summary to the paper as required: “Tsetse flies transmit human and animal African trypanosomiasis. To inform their management in heterogeneous environments, we developed a mechanistic spatio-temporal model of their population dynamics. Pest control had different outcomes depending on its spatial implementation. When homogeneously applied, refuges appeared in the coldest and temperature-stable locations. When targeting optimal locations, the effect of temperature disappeared but resurgence was high at the interface between controlled and uncontrolled zones. Differences were driven by fly capacity to disperse according to the suitability of their surroundings. This highlights the importance of identifying potential refuges through baseline data collection before starting tsetse control campaigns.”

We also shortened a bit the length of the paper as required by the system after submission, essentially removing non-essential references.

Reviewer(s)' Comments to Author

Referee: 2 - Comments to the Author(s)

Thank you for your extensive revisions. The spatial results are indeed interesting and well worth highlighting.

Thank you.

Referee: 4 - Comments to the Author(s)

I was one of the reviewers of versions of this manuscript submitted to Peers Community in Ecology and RSPB, and have already made my major comments at that time. I found the current version very strong even if it can be sometimes difficult to follow between the main text and the supplementary materials. But I am aware of the length restrictions, so I think it is fine as it is. Congratulations on the work. I am looking forward to seeing this published after all your efforts.

Thank you.

Three general comments hereafter.

1. Despite the length restriction, it would be paramount to state in the abstract what characterised the targeted locations for heterogeneous control and to clarify policy take-home message for tsetse control in a heterogeneous landscape.

The new abstract (199 words) now reads: “Spatio-temporally heterogeneous environment may lead to unexpected population dynamics. Knowledge is needed on local properties favouring population resilience at large scale. For pathogen vectors, such as tsetse flies transmitting human and animal African trypanosomiasis, this is crucial to target management strategies. We developed a

mechanistic spatio-temporal model of the age-structured population dynamics of tsetse flies, parametrized with field and laboratory data. It accounts for density- and temperature-dependence. The studied environment is heterogeneous, fragmented, and dispersal is suitability-driven. We confirmed that temperature and adult mortality have a strong impact on tsetse populations. When homogeneously increasing adult mortality, control was less effective and induced faster population recovery in the coldest and temperature-stable locations, creating refuges. To optimally select locations to control, we assessed the potential impact of treating them and their contribution to the whole population. This heterogeneous control induced a similar population decrease, with more dispersed individuals. Control efficacy was no longer related to temperature. Dispersal was responsible for refuges at the interface between controlled and uncontrolled zones, where resurgence after control was very high. The early identification of refuges, which could jeopardize control efforts, is crucial. We recommend baseline data collection to characterize the ecosystem before implementing any measures.”

2. My understanding is that mortality was increased and kept constant for the whole year of intervention, rather than implemented as discrete-time intervention waves, as one would expect interventions to be implemented. Could you justify your approach and add a couple of lines in the discussion about how relevant this approach is with regards to actual on-site intervention strategies? We added a few lines on that point in the discussion (L276-281): “We favoured spatial heterogeneity over temporal variability of control measures because it is closer to what is observed in the field. In a control programme, insecticide traps are set at a heterogeneous density based on the availability of suitable habitat, and then maintained and replaced when necessary (at least every 6 months) to maintain the control pressure [39]. Likewise, the density of sterile males to be released is proportional to the amount of suitable habitat and constant over time.”

3. It is not fully clear of the different intervention strategies are comparable in terms of “effort” deployed. Indeed, I would assume that increasing mortality by 1.7 in 50% of cells does not require the same effort than increasing mortality by 1.6 in 100% of cells. Is there an objective way of measuring the “cost”? You could add this in your discussion for perspectives, if relevant. Measuring the cost of the intervention is complex and beyond the scope of this paper (Bouyer et al. 2014). However, if we consider traps, the cost is quite proportional to the number of traps set (~5€/traps in the Niayes area) and thus the proportion of patches to be treated. So, it would actually cost twice more to treat 100% instead of 50% of the patches with traps, and it would be interesting to see the cost-effectiveness regarding the suppression rate and recovery. We added a few line at the end of the discussion (L341-345): “It (our approach) provides cues on how to spatially optimize control but could further minimize the operational burden by proposing optimal periods of intervention. Future developments should include more realistic, diverse, and customizable control scenarios, evaluated not only based on their efficacy for population reduction but also their cost in terms of on-the-ground implementation effort.”

Ref cited

Bouyer, F., M. T. Seck, A. Dicko, B. Sall, M. Lo, M. Vreysen, E. Chia, J. Bouyer, and A. Wane. 2014. Ex-ante cost-benefit analysis of tsetse eradication in the Niayes area of Senegal. *PLoS Negl. Trop. Dis.* 8: e3112.